# ControlSynth Neural ODEs: Modeling Dynamical Systems with Guaranteed Convergence

**Wenjie Mei**[*†]    **Dongzhe Zheng**[*†]    **Shihua Li**[†]

## Abstract

Neural ODEs (NODEs) are continuous-time neural networks (NNs) that can process data without the limitation of time intervals. They have advantages in learning and understanding the evolution of complex real dynamics. Many previous works have focused on NODEs in concise forms, while numerous physical systems taking straightforward forms, in fact, belong to their more complex quasi-classes, thus appealing to a class of general NODEs with high scalability and flexibility to model those systems. This, however, may result in intricate nonlinear properties. In this paper, we introduce ControlSynth Neural ODEs (CSODEs). We show that despite their highly nonlinear nature, convergence can be guaranteed via tractable linear inequalities. In the composition of CSODEs, we introduce an extra control term for learning the potential simultaneous capture of dynamics at different scales, which could be particularly useful for partial differential equation-formulated systems. Finally, we compare several representative NNs with CSODEs on important physical dynamics under the inductive biases of CSODEs, and illustrate that CSODEs have better learning and predictive abilities in these settings.

## 1 Introduction

Neural ODEs (NODEs) [4] were developed from the limiting cases of continuous recurrent networks and residual networks and exhibit non-negligible advantages in data incorporation and modeling unknown dynamics of complex systems, for instance. Their continuous nature makes them particularly beneficial to learning and predicting the dynamical behavior of complex physical systems, which are often difficult to realize due to sophisticated internal and external factors for the systems.

Starting from the introduction of NODEs, many types of variants of NODEs have been studied (see, *e.g.*, [6, 19, 17, 15]). Nevertheless, there are rare studies concerning highly scalable and flexible dynamics that also present complex nonlinear natures, bringing difficulties in their modeling and analyses. No in-detail research attention has been paid to the scalability of depth and structure in NODEs despite numerous physical systems in the real world having inscrutable dynamics and compositions. To fill in this gap, we propose ControlSynth Neural ODEs (CSODEs), in whose structure another sub-network is also incorporated for enlarging the dexterity of the composition and controlling the evolution of the state. Different from most of the existing methods and experiments, we focus on widely investigated physical models, with known state evolution under necessary conditions. Subsequently, we will show that the proposed NODEs are effective in learning and understanding those models.

Our contributions are mainly composed of the novel structure of CSODEs, their convergence guarantees, and the comparative experiments among CSODEs, several representative NODEs, and their

---

*Equal contribution. Correspondence to `wenjie.mei@seu.edu.cn` and `lsh@seu.edu.cn`

†Wenjie Mei and Shihua Li are with the School of Automation and the Key Laboratory of MCCSE of the Ministry of Education, Southeast University, Nanjing, China. Dongzhe Zheng is with the Department of Computer Science and Engineering, Shanghai Jiao Tong University, Shanghai, China.

38th Conference on Neural Information Processing Systems (NeurIPS 2024).

divisions, illustrating the beneficiality of CSODEs in the setting of prediction. The convergence conditions provide tractable solutions for constraining the learned model to a convergent one. The preliminary experiment demonstrates that our CSODEs can learn the evolution of the dynamics faster and more precisely. Also, we show that introducing sub-networks into CSODE does not impact the overall computational performance more than the other comparable NNs. Finally, we compare NODE, Augmented Neural ODE (ANODE), Second Order Neural ODE (SONODE), CSODE, and its variant in real dynamical systems. The experimental results indicate that our CSODE is beneficial to more accurate time series prediction in the systems. Our code is available online at https://github.com/ContinuumCoder/ControlSynth-Neural-ODE.

## 2 ControlSynth Neural Ordinary Differential Equations

We begin by introducing the form of CSODEs as follows:

$$\dot{x}(t) = A_0 x(t) + \sum_{j=1}^{M} A_j f_j(W_j x(t)) + g(u(t)), \quad (1)$$

where $x_t := x(t) \in \mathbb{R}^n$ is the state vector; the matrices $A.$ are with approximate dimensions; $W.$ are weight matrices; the input $u_t := u(t) \in U \subset \mathbb{R}^m$, $u \in \mathscr{L}_\infty^m$ (refer to Appendix C.1); $f_j = [f_j^1 \dots f_j^{k_j}]^\top$ $\left(f_j : \mathbb{R}^{k_j} \to \mathbb{R}^{k_j}\right)$ and $g : U \to \mathbb{R}^n$ are the functions ensuring the existence of the solutions of the neural network (NN) (1) at least locally in time, and $g = [g_1 \dots g_n]^\top$; w.l.o.g., the time $t$ is set as $t \geq 0$.

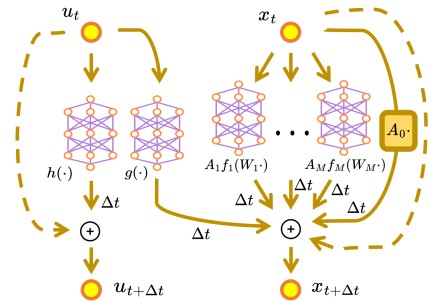

Figure 1: Schematic of the CSODEs solver, showing integration via NNs at one time step. Using the forward Euler method as an example, it shows how $u_t$ and $x_t$ evolve through the update neural function $h(\cdot)$ and NNs $g(\cdot)$, $A_1 f_1(W_1 \cdot)$, ..., $A_M f_M(W_M \cdot)$ to yield $u_{t+\Delta t}$ and $x_{t+\Delta t}$ as the next variables.

CSODEs extend the concept of Neural ODEs, which are typically expressed as $\dot{x}(t) = f(x(t))$, where $f$ is a neural network. CSODEs incorporate control inputs $u(t)$ and create a combination of subnetworks $\sum_{j=1}^{M} A_j f_j(W_j x(t))$. This formulation enhances expressiveness and adaptability to complex systems with external inputs. CSODEs provide a more universal framework and improve upon Neural ODEs by offering greater flexibility, interpretability through separated linear and nonlinear terms, and natural integration with techniques in control theory.

For simplicity, in the experiments (see Section 6), we select two common NNs as the specific examples of the function $g$: Traditional Multilayer Perceptron (MLP) and two-layer Convolutional Neural Network (CNN), and tanh as the used activation functions: a particular subclass of the functions $f_j$. Note that in the case that $g(u)$ represents an MLP, there are many different types of neurons, for example, functional neurons [21], and in CSODEs (1) there may exist $f_j^i \in L^1(X, \mathcal{A}, \mu)$, where $(X, \mathcal{A}, \mu)$ denotes a $\sigma$-finite measure space.

## 3 Related Work

**SONODEs, ANODEs, and NCDEs vs CSODEs** SONODEs [17] are particularly suitable for learning and predicting the second-order models, and ANODEs [6] are useful for cases where the state evolution does not play an important role. In contrast, many real models of second-order or even higher orders can be transformed into the common first-order but emerge numerous nonlinearities that are difficult to accurately treat by NODEs and their variants. Despite the intricate structure of CSODEs, they can be equipped with a significant convergence attribute. The experiments show that our CSODEs are more appropriate for model scaling and natural dynamics that exhibit complex nonlinear properties. Although both Neural CDEs (NCDEs) [12] and CSODEs extend NODEs through control mechanisms, their architectural philosophies differ. NCDEs introduce control through path-valued data driving the vector field, following $\dot{x}(t) = f(x(t))$. In contrast, CSODEs propose a more sophisticated structure combining multiple subnetworks with a dedicated control term $g(u(t))$. This design not only provides theoretical convergence guarantees but also enables CSODEs

to capture intricate physical dynamics through its hierarchical structure, particularly beneficial for systems exhibiting complex nonlinear behaviors that elude simpler controlled formulations.

**Convergence and Stability**   Relevant to our work, [13] studies the asymptotic stability of a specific class of NODEs as part of its fundamental property investigations on NODEs. In contrast, our study focuses on the scalability aspects related to both the number of subnetworks and the width of NNs, extending beyond the scope of the NNs in [13, 15, 16]. Furthermore, our experiments are primarily toward learning and predicting the dynamic behaviors of physical models rather than image classifications. Stability analyses have also been performed on NODE variants, such as SODEF [11] SNDEs [26], and Stable Neural Flows[14].

**Physical Information Based Dynamics**   Similar to the way CSODEs embody physical dynamics, various other models have been developed to incorporate physical information, ensuring the derivation of physically plausible results, both in discrete and continuous time. For example, [3] employs physics-informed neural networks (PINNs) (In contrast to NODEs and their variants, which mainly focus on implicitly learning dynamical systems, PINNs explicitly incorporate physical laws into the loss function. These two types of methods tackle the problem of integrating physical knowledge into deep learning models from different perspectives and are complementary research directions) to solve the Shallow Water Equations on the sphere for meteorological use, and Raissi et al. [20] propose a deep learning framework for solving forward and inverse problems involving nonlinear partial differential equations using PINNs. Shi et al. [23] and O'Leary et al. [18] utilize enhanced NODEs for learning complex behaviors in physical models, while Greydanus et al. [9] and Cranmer et al. [5] propose Hamiltonian and Lagrangian NNs, respectively, to learn physically consistent dynamics.

## 4   Theoretical Results: Convergence Analysis

Consider the CSODEs given in Equation (1), where $f_j : \mathbb{R}^{k_j} \to \mathbb{R}^{k_j}$ are nonlinear activation functions. For them, an imposed condition on $f_j^i$ (the $i$-th element of the vector-valued $f_j$) is presented as follows:

**Assumption 1** *For any $i \in \{1, \ldots, k_j\}$ and $j \in \{1, \ldots, M\}$, $sf_j^i(s) > 0$ for all $s \in \mathbb{R}\backslash\{0\}$.*

**Remark 1** *Assumption 1 applies to many activation functions, such as* tanh *and parametric ReLU. It picks up the activation functions passing through the origin and the quadrants I and III. For more explanations, the reader is referred to Appendix B.1.*

In this study, to analyze the convergence property of the NN (1), we first define the concept of *convergence*:

**Definition 1** *The model* (1) *is convergent if it admits a unique bounded solution for $t \in \mathbb{R}$ that is globally asymptotically stable (GAS).*

In order to investigate the convergence, two properties have to be satisfied, that is, the boundedness and the GAS guarantees of the solution $x^*$ for (1). In this respect, two assumptions are given as follows.

**Assumption 2** *Assume that the functions $f_j^i$ are continuous and strictly increasing for any $i \in \{1, \ldots, k_j\}$ and $j \in \{1, \ldots, M\}$.*

Assumption 2 aligns with CSODE's structure, reflecting continuity and monotonicity of activation functions. This relates to model dynamics and is satisfied by most common activations.

In the analysis of convergence, one needs to study two models in the same form but with different initial conditions and their contracting properties. To that end, along with (1), we consider the model $\dot{y}(t) = A_0 y(t) + \sum_{j=1}^{M} A_j f_j(W_j y(t)) + g(u(t))$ with the same input but different initial conditions $y(0) \in \mathbb{R}^n$. Let $\xi := y - x$. Then the corresponding error system is

$$\dot{\xi} = A_0 \xi + \sum_{j=1}^{M} A_j p_j(x, \xi), \tag{2}$$

where $p_j(x, \xi) = f_j(W_j(\xi + x)) - f_j(W_j x)$. Note that for any fixed $x \in \mathbb{R}^n$, the functions $p_j$ in the variable $\xi \in \mathbb{R}^n$ satisfy the properties formulated in Assumptions 1, 2. The following assumption is imposed for analyzing the contracting property of (2).

**Assumption 3** *Assume that there exist positive semidefinite diagonal matrices $S_0^j, S_1^j, S_2^j, S_3^{j,r}, H_0^j, H_1^j, H_2^j, H_3^{j,r}$ $(j, r \in \{1, \ldots, M\})$ with appropriate dimensions such that*

$$p_j(x,\xi)^\top p_j(x,\xi) \leq \xi^\top W_j^\top S_0^j W_j \xi + 2\xi^\top W_j^\top S_1^j p_j(x,\xi) + 2\xi^\top W_j^\top S_2^j f_j(W_j\xi)$$

$$+ 2\sum_{r=1}^{M} p_j(x,\xi)^\top W_j^\top W_j S_3^{j,r} W_r^\top W_r f_r(W_r\xi)$$

*and*

$$f_j(W_j\xi)^\top f_j(W_j\xi) \leq \xi^\top W_j^\top H_0^j W_j \xi + 2\xi^\top W_j^\top H_1^j p_j(x,\xi) + 2\xi^\top W_j^\top H_2^j f_j(W_j\xi)$$

$$+ 2\sum_{r=1}^{M} p_j(x,\xi)^\top W_j^\top W_j H_3^{j,r} W_r^\top W_r f_r(W_r\xi)$$

*for all $x, y \in \mathbb{R}^n$ and $\xi = x - y$.*

Notice that Assumption 3 is at least more relaxed than Lipschitz continuity (see Appendix B.2 for an intuitive example of activation functions satisfying Assumption 3).

**Convergence Conditions**   We are now ready to show the convergence conditions for the CSODEs (1):

**Theorem 1** *Let Assumptions 1-3 be satisfied. If there exist positive semidefinite symmetric matrices $P, \tilde{P}$; positive semidefinite diagonal matrices $\{\Lambda^j = \mathrm{diag}(\Lambda_1^j, \ldots, \Lambda_n^j)\}_{j=1}^{M}$, $\{\tilde{\Lambda}^j = \mathrm{diag}(\tilde{\Lambda}_1^j, \ldots, \tilde{\Lambda}_n^j)\}_{j=1}^{M}$, $\{\Xi^s\}_{s=0}^{M}$, $\{\Upsilon_{s,r}\}_{0 \leq s < r \leq M}$, $\{\tilde{\Upsilon}_{j,j'}\}_{j,j'=1}^{M}$, $\{\Gamma_j\}_{j=1}^{M}$, $\{\Omega_j\}_{j=1}^{M}$, $\tilde{\Xi}^0$; positive definite symmetric matrix $\Phi$; and positive scalars $\gamma, \theta$ such that the linear matrix inequalities (LMIs) in Appendix B.4 hold true. Then, a forward complete system (1) is convergent.*

Proof in Appendix C.3. Note that the used conditions on $f_j^i$ in Assumption 3 can be relaxed to "non-decreasing", which enlarges the scope of activation functions, including non-smooth functions like ReLU, then the resulting modifications for the formulations of Theorem 1 can be readily obtained, highlighting the CSODE framework's adaptability.

Those LMI conditions ensure system convergence. From an energy perspective, this indicates the error system's generalized energy (represented by the energy (or Lyapunov) function) is monotonically non-increasing, leading to convergence towards the equilibrium point: origin. These conditions can be easily verified, thanks to CSODE's structural characteristics and LMIs' highly adjustable elements.

The matrices, such as $\tilde{\Xi}^0$ and $\tilde{\Upsilon}_{j,j'}$, in the LMIs act as compensation terms balancing the effects of linear and nonlinear terms, ensuring the derivative of the energy function $\tilde{V}$ remains non-positive. Properties of $f_j$ (Assumptions 1 and 2) provide facilitation in constructing these matrices. Assumption 3 allows for non-restrictive conditions on activation functions, avoiding strong global Lipschitz continuity assumptions and providing precise local asymptotic stability characterization.

## 5   Preliminary Experiments

**Convergence and Stability**   Convergence, an important attribute showcasing a model's learning ability, refers to its capability to consistently approach the theoretical optimal solution throughout the learning process over iterations. To validate the convergence and stability of CSODEs, we design an experiment that involves learning simple spiral trajectories, chosen for their fundamental complexity which all models should ideally handle. In this experiment, we compare CSODEs and NODEs, both based on the Latent ODE structure [4]. This setup provides a fundamental baseline for assessing convergence and ensures a fair comparison, enabling each model to demonstrate its learning capabilities under comparable conditions without bias toward specific structural advantages. The Mean Absolute Error (MAE) loss function, which measures the average of the absolute differences between the estimated trajectories and the true trajectories, is used as the indicator. We train the model on 100 noisy spiral data observation points to learn this trajectory. The MAE loss values collected over training epochs consistently, plotted in Figure 2, show that the CSODE model not only converges faster but also maintains a lower error rate compared to the traditional NODE model, particularly under the constraints of limited and noisy observational data points. Our model's ability to accurately predict the target trajectory despite the noisy data illustrates its noise tolerance, a critical aspect of its robust stability.

**Generalization and Extrapolation** Figure 3 presents a visual comparison of the prediction results from the CSODE model and the traditional NODE model with the actual ground truth trajectory after various training epochs in trajectories learning experiment in the previous paragraph. It is evident from the figure that the CSODE not only learns the trajectory within the observation period more precisely in fewer training epochs but also aligns more accurately with the original trajectories beyond the observation period, which accounts for 25% of the total duration. This demonstrates the robust generalization and extrapolation capabilities of CSODEs for predicting future states, illustrating its better understanding of the underlying time-dependent structures and dynamics in the data.

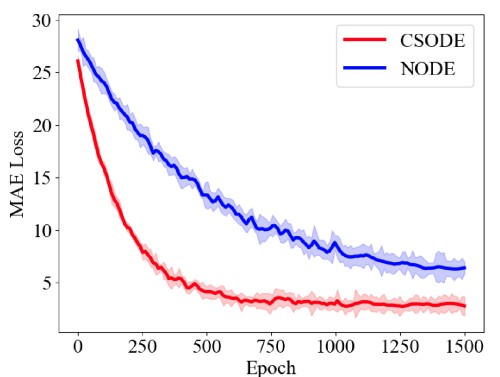

Figure 2: Comparison of Mean Absolute Error (MAE) loss reduction curves across 1500 training epochs between CSODE and NODE models until convergence.

Figure 3: Qualitative comparison of the CSODE (top) and the NODE (bottom) predictions against ground truth trajectories at 400, 600, and 800 training epochs.

**Computational Performance** To evaluate whether introducing subnetworks in CSODEs impacts the overall computational performance, we conduct experimental validations on image classification, focusing on aspects such as operational efficiency and system load.

Table 1: Performance Comparison on the MNIST

| Model | Test Err | # Params | FLOPS | Time | Mem |
|---|---|---|---|---|---|
| NODE | 0.42% | 0.22M | 2.11B | 9e-3s | 372MB |
| ANODE | 0.41% | 0.22M | 2.12B | 9e-3s | 372MB |
| SONODE | 0.41% | 0.22M | 2.11B | 9e-3s | 372MB |
| CSODE | 0.39% | 0.22M | 2.11B | 9e-3s | 372MB |

These experiments are based on the ODE-Nets framework [4], using the forward Euler method, and span 100 epochs on the MNIST dataset to compare the performance of NODEs [4], Augmented Neural ODEs (ANODEs) [6], Second Order Neural ODEs (SONODEs) [17], and CSODEs. All experiments in this study are conducted on a system equipped with an NVIDIA GeForce RTX 3080 GPU and CUDA, ensuring a consistent computational environment across all tests. Performance metrics include test error rate, number of parameters (# Params), floating-point operations per second (FLOPS), average batch processing time (Time), and peak memory usage (Mem). The results (see Table 1) show that although CSODEs theoretically involves iterative computation of $g(u)$, it does not significantly reduce computational efficiency in practice. While MNIST is a relatively simple dataset, the experimental outcomes preliminarily confirm the benefits of incorporating a learnable function in enhancing the performance of implicit networks, effectively maintaining efficiency, and alleviating concerns over the potential trade-off between expressive capability and computational efficiency of the networks.

## 6 Complex Systems Time Series Prediction Experiments

In this section, we experimentally analyze the performance of CSODEs, based on NODEs, compared to other models based on NODEs and traditional time series forecasting methods in extrapolative time series prediction for complex systems. We select challenging physical dynamical systems from neuroscience (Hindmarsh-Rose model [10]), chemistry and biology (Reaction-Diffusion model [8]), and geophysics and fluid dynamics (Shallow Water Equations [25]), which exhibit rich spatiotemporal variations over long time scales.

## 6.1 Model Architectures for Experiments

To comprehensively evaluate the performance of different model architectures in dynamic system modeling, this study compares several representative NODEs-based models, including the traditional NODEs [4], ANODEs [6], SONODEs [17], and our proposed CSODEs. Additionally, these models are compared with traditional MLPs [22] and Recurrent Neural Networks (RNNs) [7].

Notable that while CSODE can be integrated with the Latent ODE framework, as demonstrated in our preliminary experiments (in Section 5), we opted to conduct our main experiments without this integration. This decision was driven by our observation that NODE itself effectively models long-sequence dynamical systems, allowing us to evaluate CSODE's performance more directly by eliminating potential influences from additional Latent ODE components like encoders and decoders.

All models within our experiment that are based on the NODE framework utilize a uniform MLP to parameterize the core dynamical system $\dot{y} = f(y)$. These models employ the forward Euler method for integration, though other numerical solvers like Dopri5 are also viable options (detailed performance comparisons of different solvers can be found in Appendix I.1). The MLP serves as the update function, iterating to act on the evolution of dynamic system states. Specifically, the MLP receives the current state as input and outputs the next state, thereby modeling the change in state over time. Meanwhile, the RNN employs the same structure as the MLP, using current input and the previous timestep's hidden state to update the current timestep's hidden state. To ensure a fair comparison, we finely align the number of parameters across all models, including the parameterization of the core dynamical system and other components (such as the subnetwork in the CSODE model and the augmented dimensions in the ANODE model). All designs ensure that the total number of parameters remains consistent within $\pm 1\%$.

The unique feature of CSODEs lies in the introduction of an additional subnetwork to model the control term $g(u(t))$, extending the original dynamical system. We employ a unified auxiliary NN to model the changing rate of $g(u(t))$, with the initial value $g(u_0)$ set to be the same as $y_0$. These subnetwork structures are similar to the MLP or use simplified single-layer networks. We also introduce the CSODE-Adapt variant, replacing the function representing the changing rate of the control term with a network consisting of two convolutional layers, to explore the scalability and flexibility of CSODEs. Notably, for fair comparison of fundamental architectures, we used the standard Adam optimizer in our main experiments, though we note that alternative optimizers like L-BFGS could further enhance CSODE's performance (see Appendix I.2 for more details).

At the current research forefront, some researchers have proposed integrating NODEs with Transformer layers (TLODEs) [27]. TLODEs can be considered a special case of CSODEs, where the Transformer layers implement the function $f(\cdot)$. Building on TLODEs, we introduce the control term $g(\cdot)$, creating a more complete CSODEs structure with Transformer layers (CSTLODEs). Considering the widespread application of Transformers in time-series prediction, we conduct comparative experiments between the Transformer, TLODE, and CSTLODE models. However, due to significant architectural differences between Transformers and MLPs and RNNs, directly incorporating them into the main experiment might introduce additional confounding factors, deviating from the theoretical discussion of general NODEs. Therefore, to maintain the focus and clarity of the main text, we have placed the experimental results and discussions related to TLODEs and CSTLODEs in Appendix F.

## 6.2 Experimental Tasks and Datasets Description

In this subsection, we detail the experimental tasks and datasets used to explore the application of NNs in simulating various dynamic systems. For more comprehensive simulation and experimental details, see Appendix D.

**Modeling Hindmarsh-Rose Neuron Dynamics** In this task, we explore the application of NNs in simulating the Hindmarsh-Rose neuron model [10]. We validate their potential in simulating complex neuronal dynamics and assess their prospects in broader neuroscience and biophysical research. The Hindmarsh-Rose model is widely used to describe neuronal dynamic behavior, particularly suitable for studying neuronal firing behavior and chaotic phenomena. This model describes the evolution of the neuron's voltage and recovery variables over time with the following three coupled nonlinear differential equations:

$$\frac{dx}{dt} = y - ax^3 + bx^2 - z + I, \quad \frac{dy}{dt} = c - dx^2 - y, \quad \frac{dz}{dt} = r(s(x - x_0) - z), \quad (3)$$

where $x$ represents the membrane potential, $y$ represents the recovery variable associated with the membrane potential, and $z$ is the variable for adaptive current. The parameters $a$, $b$, $c$, $d$, $r$, $s$, $x_0$, and $I$ determine the neuron's nonlinear firing behavior, voltage response curve, recovery process rate, and adaptive current, simulating the voltage changes and behavior of the neuron.

We conduct 1000 simulations, each lasting 20 seconds. In the first 10 seconds of each simulation, we collect 1500 data points reflecting the changes in the neuron's three-dimensional coordinates $(x, y, z)$. We divide these data points into 50 sequences, each containing 30 time points. Our goal is to use these sequences to train an NN to predict the dynamic changes over the next 10 seconds, specifically the sequences of the next 30 time points.

**Modeling Reaction-Diffusion System**    In this task, we employ NN models to simulate the Gray-Scott equations [8], a Reaction-Diffusion system widely used to describe chemical reactions and diffusion processes. This system is particularly significant in pattern formation, biological tissue, and self-organizing chemical processes, holding important theoretical and practical implications. This task allows us to explore the potential applications of these models in environmental and bioengineering contexts. The model describes the interactions and spatial diffusion of two chemical substances, $U$ and $V$, through a set of partial differential equations (PDEs):

$$\frac{\partial U}{\partial t} = D_U \nabla^2 U - UV^2 + j(1 - U), \quad \frac{\partial V}{\partial t} = D_V \nabla^2 V + UV^2 - (j + k)V, \quad (4)$$

where $D_U$ and $D_V$ are the diffusion coefficients, and $j$ and $k$ are reaction rate parameters.

We conducted 1000 simulations with periodic boundary conditions [1] for 150 seconds, each randomizing initial conditions and diffusion coefficients $D_U$ and $D_V$, and used data from the first 100 seconds. These 100 seconds of data were divided into 100 sequences, each spanning 25 seconds, to train the NNs. Our goal is to utilize the training results to predict chemical dynamics over the next 50 seconds.

**Modeling Shallow Water Equations**    In this task, we explore the application potential of NODE systems in simulating the Shallow Water Equations [25], which describe the horizontal flow and surface behavior of liquids in confined spaces. These equations have significant physical relevance in fields such as hydrology and environmental engineering, especially in flood simulations and wave dynamics analysis. Mathematically, the Shallow Water Equations are expressed through a set of nonlinear PDEs that represent changes in water depth ($h$) and flow velocity ($\mathbf{u}$), incorporating the principles of continuity and conservation of momentum, with gravity $g$, as follows:

$$\frac{\partial h}{\partial t} + \nabla \cdot (h\mathbf{u}) = 0, \quad \frac{\partial (h\mathbf{u})}{\partial t} + \nabla \cdot (h\mathbf{u} \otimes \mathbf{u}) + \frac{1}{2}g\nabla h^2 = 0. \quad (5)$$

We conduct 1000 periodic boundary condition wave propagation simulations [1] lasting 7 seconds, generating 1500 data points in the first 4.7 seconds to describe water wave depth variations. After normalizing the data using a sliding window, we segment it into 100 sequences, each containing 15 points, to train a global water surface dynamics model to predict water wave depth changes during the remaining 2.3 seconds of the simulation.

### 6.3   Metrics for Assessing Prediction Accuracy

We employ the following metrics to compare time series predictions and ground truth:

**Mean Squared Error (MSE):** Calculates the mean of squared differences between predicted and actual values, sensitive to large errors. Used in all three tasks.

**Mean Absolute Error (MAE):** Measures the average absolute difference between predicted and actual values, robust to noise. Used in all three tasks.

**$R^2$ Score:** Quantifies the proportion of variance in predictions relative to actual data, reflecting explanatory power and accuracy. Used in the Hindmarsh-Rose model's 3D time series prediction.

**Chamfer Distance (CD):** Measures the average shortest distance between points in two sets, emphasizing spatial structure matching accuracy [2]. Used to compare predicted and ground truth physical field sequences in Reaction-Diffusion systems and Shallow Water Equations tasks.

## 6.4  Experimental Results and Analysis

Experiments are conducted ten times for each task, and the average metrics are presented in Table 2, the reader is referred to Appendix E for more statistical details. NNs in Group B, based on NODE and its variants, outperform traditional models in Group A, such as MLP and RNN, in predicting complex physical dynamic systems. Traditional models struggle with complex time dependencies and nonlinear dynamics, while models based on differential equations demonstrate greater adaptability and accuracy.

Table 2: The table categorizes NN models for time-series prediction in complex dynamical systems into three groups: Group A includes traditional models like MLP and RNN; Group B features ODE-based models including the NODE, ANODE, SONODE, and CSODE models; Group C presents CSODE-Adapt with convolutional layers. The best-performing models are highlighted in blue and the second-best in brown.

| Group | Model | Hindmarsh-Rose | | | Reaction-Diffusion | | | Shallow Water | | |
|---|---|---|---|---|---|---|---|---|---|---|
| | | MSE | MAE | $R^2$ | MSE | MAE | CD | MSE | MAE | CD |
| A | MLP | 2.394 | 1.033 | 0.241 | 7.431e-2 | 0.398 | 7.745 | 0.135 | 0.35 | 3.17 |
| | RNN | 1.975 | 0.871 | 0.356 | 2.376e-2 | 0.143 | 3.115 | 0.103 | 0.26 | 3.06 |
| B | NODE | 1.551 | 0.682 | 0.515 | 1.134e-2 | 0.081 | 1.311 | 0.071 | 0.22 | 2.89 |
| | ANODE | 0.745 | 0.586 | 0.637 | 1.095e-2 | 0.076 | 1.392 | 0.067 | 0.19 | 2.88 |
| | SONODE | 0.739 | 0.561 | 0.611 | 8.145e-3 | 0.056 | 1.315 | 0.065 | 0.19 | 2.83 |
| | CSODE (Ours) | 0.783 | 0.470 | 0.758 | 6.365e-3 | 0.058 | 0.939 | 0.041 | 0.16 | 2.27 |
| C | CSODE-Adapt (Ours) | 0.408 | 0.370 | 0.887 | 5.635e-3 | 0.035 | 0.837 | 0.031 | 0.15 | 1.89 |

Within Group B, the CSODE model surpasses other models due to its control elements that enhance precision and adaptability to changes in initial conditions and system parameters. Figures 4 and 5 show that CSODEs accurately reflect complex dynamic system details through qualitative results, with more results available in Appendix G.

Moreover, the CSODE-Adapt model (Group C) integrates convolutional layers, enhancing its applicability and effectiveness, particularly in dynamic systems with significant spatial features, such as Reaction-Diffusion systems. This model performs better than all others, highlighting the flexibility and highly customizable structure of the CSODEs and its advantages and potential in predicting complex physical dynamic systems.

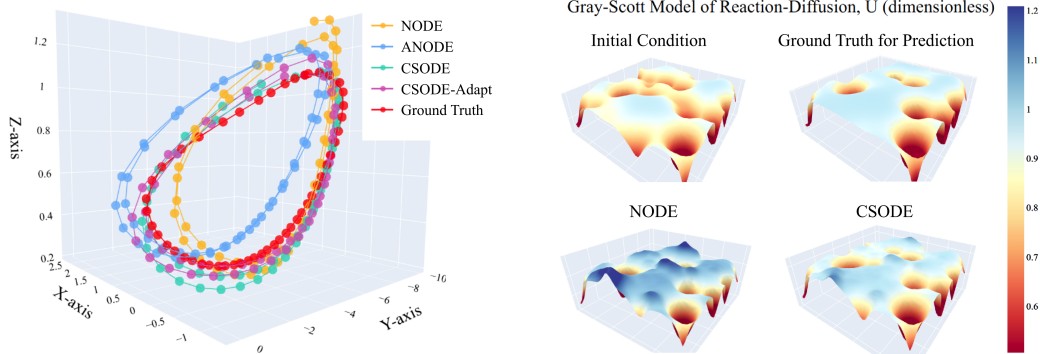

Figure 4: Qualitative results of Hindmarsh-Rose model prediction

Figure 5: Qualitative results of Reaction-Diffusion system prediction

## 6.5  Comparison with Observation-aware Baselines

We conducted supplementary experiments (CharacterTrajectories and PhysioNet Sepsis Prediction) with CSODE-Adapt following the experimental setup in [12], maintaining similar network structures,

parameter counts, and optimization methods. Overall, CSODE performs slightly worse than Neural CDE in irregular observation experiments but better in other time-series-related tasks. We have also performed our main experiments for neural CDE. The corresponding results are shown as follows:

Table 3: Performance comparison with observation-aware neural networks. Group A includes observation-aware baselines like ODE-RNN and Neural CDE; Group B contains our base model CSODE; Group C showcases CSODE-Adapt with enhanced observation mechanisms. Best performing models are marked in blue, second-best in brown.

| Metric / Model Group | Neural CDE A | ODE-RNN A | CSODE B | CSODE-Adapt C |
|---|---|---|---|---|
| **CharacterTrajectories (Test Acc.)** | | | | |
| 30% Missing | 97.8% | 96.8% | 97.3% | 97.1% |
| 50% Missing | 98.2% | 96.5% | 97.8% | 97.6% |
| 70% Missing | 97.2% | 95.9% | 96.8% | 96.5% |
| **PhysioNet Sepsis Prediction (AUC)** | | | | |
| w/o OI | 0.865 | 0.855 | 0.868 | 0.871 |
| w/ OI | 0.885 | 0.870 | 0.881 | 0.883 |
| **Reaction-Diffusion** | | | | |
| MSE | 7.1e-3 | 7.5e-3 | 7.0e-3 | 6.8e-3 |
| MAE | 0.60 | 0.62 | 0.59 | 0.58 |
| CD | 0.945 | 0.950 | 0.942 | 0.940 |

As shown in Table 3, for the **CharacterTrajectories** task with irregular observations, Neural CDE achieves the best performance across all missing data ratios, followed by CSODE and then ODE-RNN. In the **PhysioNet Sepsis Prediction** task, without considering observation intensity (OI), CSODE-Adapt achieves the highest AUC value of 0.871, while with OI consideration, Neural CDE performs best with an AUC of 0.885, followed closely by CSODE-Adapt and CSODE. For the **Reaction-Diffusion** modeling task, CSODE-Adapt demonstrates superior performance across all metrics (MSE, MAE, CD), while CSODE and Neural CDE show comparable performance, both outperforming ODE-RNN. Overall, while Neural CDE exhibits advantages in handling irregular observations, CSODE-Adapt shows competitive or superior performance in tasks requiring complex dynamic system modeling and clinical prediction, demonstrating its effectiveness as a general-purpose time series modeling tool.

# 7    Model Scaling Experiment

In the experiments above, CSODEs demonstrate significant superiority over traditional NODEs and their variants, under the maintenance of the same number of parameters and architectural configuration. Based on these findings, our scaling experiments focus primarily on exploring the scalability and architectural robustness of CSODEs, without further comparison to other models.

We also observe changes in system performance after scaling CSODEs. To maintain consistency in the experiments, each sub-network is configured with two dense layers. We select the Reaction-Diffusion task in Section 6 as an example to explore the impact of increasing the number of sub-networks and the width of each sub-network on system performance. Specifically, the network widths, which refer to the number of hidden dimensions in the dense layers, are set at 128, 256, 512, 1024, and 2048. The number of sub-networks, equivalent to $M$ in NNs (1), is set at 1, 2, 3, 4, and 5.

The experimental design varies network width and number of sub-networks. We employ a learning rate formula: learning rate $= \frac{k}{W \times \sqrt{N}}$, where $k$ is a constant, $W$ is the network width, and $N$ is the number of sub-networks. This adjusts the learning rate based on width and moderates it for sub-network count to handle complexity. For instance, with a width of 1024 and three sub-networks, the learning rate is $\frac{0.1}{1024 \times \sqrt{3}}$. We use the Adam optimizer for training.

In terms of overall performance, the heatmap in Figure 6 shows that under the CSODEs, increasing the network width and number of subnetworks results in stable and enhanced overall performance. Additionally, the scatter plot demonstrates that increasing the number of subnetworks significantly improves the model's generalization ability, with training and validation performance showing a stronger correlation. For further details on comparative experiments, the model's stability, and convergence despite the increase in the number of subnetworks and width, refer to Appendix H.

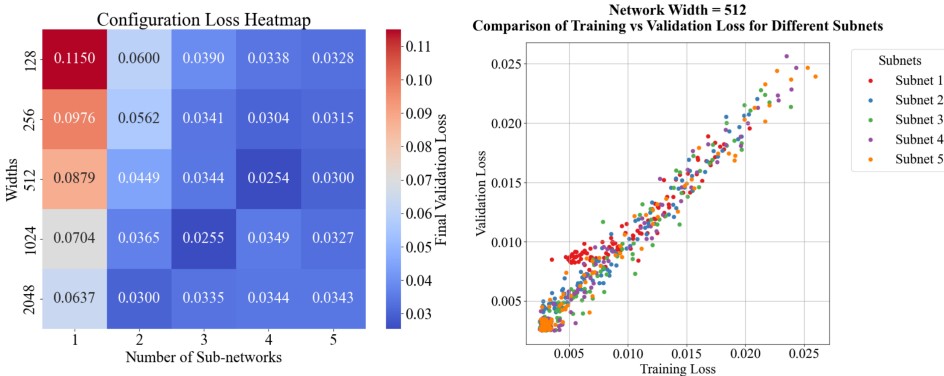

Figure 6: Performance comparison of CSODE models with varying numbers of sub-networks. The scatter plot visualizes the performance trajectory of CSODE models during training, where each point represents the training loss (x-axis) and validation loss (y-axis) at a specific epoch. Models with 1-5 sub-networks are compared, each depicted in a distinct color while maintaining a fixed network width of 512 neurons. Points clustering near the bottom-left corner indicate superior model performance, while their distribution relative to the diagonal reveals the balance between training and validation performance.

## 8   Conclusion

In this work, we analyzed the learning and predicting abilities of Neural ODEs (NODEs). A class of new NODEs: ControlSynth Neural ODEs (CSODEs) was proposed, which has a complex structure but high scalability and dexterity. We started by investigating the convergence property of CSODEs and comparing them with traditional NODEs in the context of generalization, extrapolation, and computational performance. We also used a variety of representative NODE models and CSODEs to model several important real physical dynamics and compared their prediction accuracy.

We presented that although the control subjects (NODEs, Augmented Neural ODEs (ANODEs), and Second Order NODEs (SONODEs)) do not have the physics information-based inductive biases specifically owned by our CSODEs, they can learn and understand complex dynamics in practice. In particular, SONODEs own inductive biases for second-order ODE-formulated dynamics, while the ones of CSODEs mainly are for first-order models with high nonlinear natures, scalability, and flexibility that belong to a broad class of real systems. The experimental results on dynamical systems governed by the Hindmarsh-Rose Model, Reaction-Diffusion Model, and Shallow Water Equations preliminarily demonstrate the superiority of our ControlSynth ODEs (CSODEs) in learning and predicting highly nonlinear dynamics, even when represented as partial differential equations.

**Limitations and Future Work**   The effectiveness of inductive biases of CSODEs varies which, depending on the specific application scenarios, may not be preferable; in the evolution of partial differential equations, there often exists mutual interference between different scales (*e.g.*, spatial and temporal scales), which, however, is approximately learned by CSODEs. We believe this work could provide promising avenues for future studies, including: For enlarging the use scope of inductive biases of CSODEs in complex dynamics and reflecting the mutual intervention between scales, one can consider a more general NODE: $\dot{x} = f(x, u)$, with guaranteed stability and convergence. This allows a greater scope of dynamics and thus may prompt the improvement of the accuracy of modeling and predicting systems with more complex structures and behaviors. Furthermore, in practice, CSODEs are more sensitive to learning rate selections due to their more complex architectures. Our preliminary investigation in Appendix I.3 reveals the significant impact of hyperparameter adjustments on model performance. Building upon these initial findings, future research will examine this sensitivity more thoroughly and consider methods like adaptive learning rate adjustment and model simplification to address it. Finally, this work maintained standard optimization settings for fair comparison, future research could explore specialized training algorithms that leverage CSODE's structural properties and theoretical foundations.

## Acknowledgments

This work is supported by the National Natural Science Foundation of China (NSFC) under grant 62403125, the Natural Science Foundation of Jiangsu Province, and the Fundamental Research Funds for the Central Universities under grants 2242024k30037 and 2242024k30038.

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

# A   Broader Impact

The introduction of ControlSynth Neural ODEs (CSODEs) represents an advancement in the field of deep learning and its applications in physics, engineering, and robotics. CSODEs offer a highly scalable and flexible framework for modeling complex nonlinear dynamical systems, enabling the learning and prediction of intricate behaviors from data, which has implications across various domains.

In the realm of physics, CSODEs provide a powerful tool for studying complex physical processes in areas such as meteorology, fluid dynamics, and materials science. By leveraging their highly expandable network structure and control term, CSODEs may capture the dynamics of systems with elusive internal and external factors, overcoming the limitations of traditional mathematical modeling approaches. This opens up new avenues for understanding and predicting the behavior of these complex systems.

Moreover, the theoretical convergence guarantees of CSODEs lay a solid foundation for applications of neural networks in physical modeling. Despite their high nonlinearity, the convergence of CSODEs can be ensured through tractable linear inequalities. This enhances the interpretability and reliability of neural network models, facilitating their deployment in engineering and safety-critical domains where provable convergence is crucial.

The control term introduced in CSODEs also enables the learning of multi-scale dynamics, which is particularly relevant for systems described by partial differential equations. By simultaneously capturing dynamics at different spatial and temporal scales, CSODEs expand the practicality of neural networks to multi-scale physical problems.

The performance of CSODEs in learning and predicting the dynamics of physical systems, as demonstrated through the comparative experiments, highlights their alignment with physical laws and their ability to effectively uncover underlying patterns in data. This motivates the incorporation of appropriate inductive biases when designing neural network architectures for scientific and engineering applications.

In the field of robotics, CSODEs have the potential to change the design and control of robotic systems. They can be applied to disturbance observation, pose estimation, and the design of control strategies. By learning the dynamics of disturbances, CSODEs enable robots to accurately estimate their states and make appropriate compensations in the face of environmental perturbations. For pose estimation, CSODEs can learn the pose dynamics directly from sensor data, establishing adaptive pose estimators that enhance accuracy and robustness in complex environments.

# B   Further Technical Details for Section 4

## B.1   Further Implications of Assumption 1

With a reordering of nonlinearities and their decomposition, there exists an index $\omega \in \{0, \ldots, M\}$ such that for all $s \in \{1, \ldots, \omega\}$ and $i \in \{1, \ldots, k_s\}$, $\lim_{\nu \to \pm\infty} f_s^i(\nu) = \pm\infty$. Also, there exists $\zeta \in \{\omega, \ldots, M\}$ such that for all $s \in \{1, \ldots, \zeta\}, i \in \{1, \ldots, k_s\}$, we have $\lim_{\nu \to \pm\infty} \int_0^\nu f_s^i(r)dr = +\infty$. Here, $\omega = 0$ implies that all nonlinearities are bounded. The sets with the upper bound smaller than the lower bound are regarded as empty sets, $e.g.$, $s \in \{1, 0\} = \emptyset$ in the case that $\omega = 0$.

## B.2   An Example of Activation Functions Satisfying Assumption 3

In practice, Assumption 3 can be verified based on chosen $f_j$ ($e.g.$, ReLU, Sigmoid, $\tanh$). For example, for the $\tanh$ function,
$$f_j(x) = \frac{e^x - e^{-x}}{e^x + e^{-x}}, \quad |f_j(x)| < 1, \quad |f_j(x) - f_j(y)| \leq |x - y|.$$
It is Lipschitz continuous with constant 1 and satisfies:
$$p_j(x, \xi)^\top p_j(x, \xi) \leq \xi^\top W_j^\top W_j \xi, \quad f_j(W_j\xi)^\top f_j(W_j\xi) \leq \xi^\top W_j^\top W_j \xi$$
In this case, we can choose:
$$S_0^j = H_0^j = I, \quad S_1^j = S_2^j = S_3^{j,r} = H_1^j = H_2^j = H_3^{j,r} = 0.$$
If $S_1^j, S_2^j, S_3^{j,r}, H_1^j, H_2^j, H_3^{j,r} > 0$, the restrictions are relaxed, verifying that Assumption 3 is less strict than Lipschitz continuity.

## B.3   Verification Experiment on Assumption 3

To verify the actual existence of matrices in Assumption 3, we designed a supplementary experiment to validate the trained CSODE models, involving three dynamical system time series prediction tasks in the main

experiments. Retrained CSODE models (3 fully-connected layers, 128 units, Softplus; Adam optimizer, 500 epochs, learning rate $=10^{-3}$, batch=64, MSE loss). We randomly selected 500 sample pairs $(x^i, y^i)$ from each task's test set, calculated the difference $\xi^i = y^i - f(x^i)$ where $f(\cdot)$ is the trained CSODE model, and used YALMIP to define optimization variables and constraints. We constructed 500 matrix inequalities $(\xi^i)^\top P \xi^i \leq 0$ and added the semi-definite constraint $P = P^\top \geq 0$. Using the Sedumi solver (precision $10^{-6}$, max 5000 iterations), we solved the optimization problem and recorded the solving time and iteration count. We then performed individual validation for each sample pair to calculate the satisfiability ratio of Assumption 3.

**Experimental Results (average of 3 independent experiments)**: For the Hindmarsh-Rose model, solving time was 15.3 minutes with 2684 iterations and 99.8% (499/500) satisfiability. The Reaction-Diffusion System task took 18.7 minutes with 3,217 iterations and 99.6% (498/500) satisfiability. The Shallow WaterEquations required 24.2 minutes with 4,105 iterations and 99.4% (497/500) satisfiability. The solver quickly found solutions satisfying Assumption 3, with >99% satisfiability across tasks, supporting our theoretical assumptions for trained models.

### B.4 Formulations of Matrix Inequalities in Theorem 1

The linear matrix inequalities in Theorem 1 are formulated as follows.

$$P + \sum_{j=1}^{\zeta} \Lambda^j > 0; \quad Q = Q^\top \leq 0; \quad \Xi^0 + \sum_{j=1}^{M} \Upsilon_{0,j} + \sum_{s=1}^{\omega} \Xi^s + \sum_{s=1}^{\omega} \sum_{r=s+1}^{\omega} \Upsilon_{s,r} > 0. \tag{6}$$

$$\tilde{P} + \sum_{j=1}^{\zeta} \tilde{\Lambda}^j > 0; \tilde{Q} = \tilde{Q}^\top \leq 0; \quad \tilde{\Xi}^0 - W_j^\top \left( \gamma \sum_{j=1}^{M} S_0^j + \theta \sum_{j=1}^{M} H_0^j \right) W_j \geq 0; \quad \Gamma_j - \gamma S_1^j - \theta H_1^j \geq 0;$$

$$\Omega_j - \gamma S_2^j - \theta H_2^j \geq 0; \quad \tilde{\Upsilon}_{j,r} - \gamma S_3^{j,r} - \theta H_3^{j,r} \geq 0;$$

$$\tilde{\Xi}^0 - W_j^\top \left( \gamma \sum_{j=1}^{M} S_0^j + \theta \sum_{j=1}^{M} H_0^j \right) W_j + \sum_{j=1}^{M} \left( \Gamma_j - \gamma S_1^j - \theta H_1^j + \Omega_j - \gamma S_2^j - \theta H_2^j \right)$$

$$+ \sum_{j=1}^{M} \sum_{r=1}^{M} \tilde{\Upsilon}_{j,r} - \gamma S_3^{j,r} - \theta H_3^{j,r} > 0, \tag{7}$$

where

$$Q_{1,1} = A_0^\top P + P A_0 + \Xi^0; \quad Q_{j+1,j+1} = A_j^\top W_j^\top \Lambda^j + \Lambda^j W_j A_j + \Xi^j;$$

$$Q_{1,j+1} = P A_j + A_0^\top W_j^\top \Lambda^j + W_j^\top \Upsilon_{0,j}; \quad Q_{s+1,r+1} = A_s^\top W_r^\top \Lambda^r + \Lambda^s W_s A_r + W_s^\top W_s \Upsilon_{s,r} W_r^\top W_r;$$

$$Q_{1,M+2} = P; \quad Q_{M+2,M+2} = -\Phi; \quad Q_{j+1,M+2} = \Lambda^j W_j.$$

$$\tilde{Q}_{1,1} = A_0^\top P + P A_0 + \tilde{\Xi}^0; \quad \tilde{Q}_{2,2} = -\gamma I; \quad \tilde{Q}_{1,2} = PA + \Gamma; \quad \tilde{Q}_{1,3} = A_0^\top \Delta + \Omega;$$

$$\tilde{Q}_{2,3} = A^\top \Delta + \tilde{\Upsilon}; \quad \tilde{Q}_{3,3} = -\theta I; \quad A = \begin{bmatrix} A_1 & \dots & A_M \end{bmatrix}; \quad \Gamma = \begin{bmatrix} W_1^\top \Gamma_1 & \dots & W_M^\top \Gamma_M \end{bmatrix};$$

$$\Delta = \begin{bmatrix} W_1^\top \tilde{\Lambda}^1 & \dots & W_M^\top \tilde{\Lambda}^M \end{bmatrix}; \Omega = \begin{bmatrix} W_1^\top \Omega_1 & \dots & W_M^\top \Omega_M \end{bmatrix}; \tilde{\Upsilon} = (W_j^\top W_j \tilde{\Upsilon}_{j,r} W_r^\top W_r)_{j,r=1}^M.$$

## C  Notation, Definitions, and Proof

### C.1  Related Notation

The symbol $\mathbb{R}$ represents the set of real numbers, $\mathbb{R}_+ = \{\ell \in \mathbb{R} : \ell \geq 0\}$, and $\mathbb{R}^n$ denotes the vector space of $n$-tuple of real numbers. The transpose of a matrix $A \in \mathbb{R}^{n \times n}$ is denoted by $A^\top$. Let $I$ stand for the identity matrix. The symbol $\|\cdot\|$ refers to the Euclidean norm on $\mathbb{R}^n$.

For a Lebesgue measurable function $u: \mathbb{R} \to \mathbb{R}^q$, define the norm $\|u\|_{(t_1, t_2)} = \operatorname{ess\,sup}_{t \in (t_1, t_2)} \|u(t)\|$ for $(t_1, t_2) \subseteq \mathbb{R}$. We denote by $\mathscr{L}_\infty^q$ the space of functions $u$ with $\|u\|_\infty := \|u\|_{(-\infty, +\infty)} < +\infty$.

A continuous function $\alpha : \mathbb{R}_+ \to \mathbb{R}_+$ belongs to class $\mathscr{K}$ if it is strictly increasing and $\alpha(0) = 0$, and $\mathscr{K}_\infty$ means that $\alpha$ is also unbounded. A continuous function $\beta : \mathbb{R}_+ \times \mathbb{R}_+ \to \mathbb{R}_+$ belongs to class $\mathscr{K}\mathscr{L}$ if $\beta(\cdot, r) \in \mathscr{K}$ and $\beta(r, \cdot)$ is a decreasing to zero function for any fixed $r > 0$.

### C.2  Related Definitions

**Definition 2** *The system* (1) *is called* forward complete *if for all $x_0 \in \mathbb{R}^n$ and $u \in \mathscr{L}_\infty^m$, the solution $x(t, x_0, u)$ is uniquely defined for all $t \geq 0$.*

**Definition 3** *A forward complete system* (1) *with the output* $y(t) := \xi(t)$ *is said to be*

1. input-to-state stable *(ISS) if there exist* $\beta \in \mathscr{KL}$ *and* $\gamma \in \mathscr{K}$ *such that*

$$\|x(t, x_0, u)\| \leq \beta(\|x_0\|, t) + \gamma(\|u\|_\infty), \quad \forall t \geq 0$$

*for any* $x(0) = x_0 \in \mathbb{R}^n$ *and* $u \in \mathscr{L}_\infty^m$.

2. state-independent input-to-output stable *(SIIOS) if there exist* $\beta \in \mathscr{KL}, \gamma \in \mathscr{K}$ *such that*

$$\|y(t, x_0, u)\| \leq \beta(\|\xi(0)\|, t) + \gamma(\|u\|_\infty), \quad \forall t \geq 0$$

*for any* $x_0 \in \mathbb{R}^n$ *and* $u \in \mathscr{L}_\infty^m$.

## C.3   Proof of the Convergence Theorem

**Proof of Theorem 1** *The proof developments are divided into two main parts: ISS and Global Asymptotic Stability (GAS) guarantees, as shown in the sequel.*

**ISS analysis of** (1) **for boundedness**   *Consider a Lyapunov function*

$$V(x) = x^\top P x + 2 \sum_{j=1}^{M} \sum_{i=1}^{k_j} \Lambda_i^j \int_0^{W_j^i x} f_j^i(s) ds, \tag{8}$$

*where the vector* $W_j^i$ *is the* $i$-*th row of the matrix* $W_j$. *It is positive definite and radially unbounded due to Finsler's Lemma under the condition* (6) *and Assumption 1. Then, taking the derivative of* $V(x)$, *one has*

$$
\begin{aligned}
\dot{V} &= \dot{x}^\top P x + x^\top P \dot{x} + 2 \sum_{j=1}^{M} \dot{x}^\top W_j^\top \Lambda^j f_j(W_j x) \\
&= x^\top \left( A_0^\top P + P A_0 \right) x + \left( \sum_{j=1}^{M} f_j(W_j x)^\top A_j^\top \right) P x + x^\top P \sum_{j=1}^{M} A_j f_j(W_j x) + 2 x^\top P g(u) \\
&\quad + 2 \sum_{j=1}^{M} \left( x^\top A_0^\top W_j^\top \Lambda^j f_j(W_j x) + g(u)^\top W_j^\top \Lambda^j f_j(W_j x) \right. \\
&\quad + \left. \left( \sum_{s=1}^{M} f_s(W_s x)^\top A_s^\top \right) W_j^\top \Lambda^j f_j(W_j x) \right).
\end{aligned}
$$

*Therefore, under the condition* (6), *we obtain*

$$
\begin{aligned}
\dot{V} &= \begin{bmatrix} x \\ f_1(W_1 x) \\ \vdots \\ f_M(W_M x) \\ g(u) \end{bmatrix}^\top Q \begin{bmatrix} x \\ f_1(W_1 x) \\ \vdots \\ f_M(W_M x) \\ g(u) \end{bmatrix} - x^\top \Xi^0 x \\
&\quad - \sum_{j=1}^{M} f_j(W_j x)^\top \Xi^j f_j(W_j x) - 2 \sum_{j=1}^{M} x^\top W_j^\top \Upsilon_{0,j} f_j(W_j x) \\
&\quad - 2 \sum_{s=1}^{M-1} \sum_{r=s+1}^{M} f_s(W_s x)^\top W_s^\top W_s \Upsilon_{s,r} W_r^\top W_r f_r(W_r x) + g(u)^\top \Phi g(u) \\
&\leq -x^\top \Xi^0 x - \sum_{j=1}^{M} f_j(W_j x)^\top \Xi^j f_j(W_j x) - 2 \sum_{j=1}^{M} x^\top W_j^\top \Upsilon_{0,j} f_j(W_j x) \\
&\quad - 2 \sum_{s=1}^{M-1} \sum_{r=s+1}^{M} f_s(W_s x)^\top W_s^\top W_s \Upsilon_{s,r} W_r^\top W_r f_r(W_r x) + g(u)^\top \Phi g(u) \\
&\leq -\alpha(V) + g(u)^\top \Phi g(u),
\end{aligned}
$$

*for a function* $\alpha \in \mathscr{K}_\infty$. *Under Theorem 1 of [24], one can verify the first condition of the ISS property due to the form of* $V$, *and the second relation can be recovered via* $V \geq \alpha^{-1}\left(2 g(u)^\top \Phi g(u)\right) \Rightarrow \dot{V} \leq -\frac{1}{2}\alpha(V)$. *This means that the ISS property of the NN* (1) *is guaranteed, and so is the boundedness of its solution.*

**Global attracting solution of the error dynamics** (2)   *Consider another positive definite and radial unbounded (for the variable ξ) function*

$$\tilde{V}(\xi) = \xi^\top \tilde{P}\xi + 2\sum_{j=1}^{M}\sum_{i=1}^{k_j}\tilde{\Lambda}_i^j \int_0^{W_j^i\xi} f_j^i(s)ds.$$

*Similarly, taking the time derivative of $\tilde{V}$:*

$$\dot{\tilde{V}} = \begin{bmatrix} \xi \\ p_1(x,\xi) \\ \vdots \\ p_M(x,\xi) \\ f_1(W_1\xi) \\ \vdots \\ f_M(W_M\xi) \end{bmatrix}^\top \tilde{Q} \begin{bmatrix} \xi \\ p_1(x,\xi) \\ \vdots \\ p_M(x,\xi) \\ f_1(W_1\xi) \\ \vdots \\ f_M(W_M\xi) \end{bmatrix}$$

$$+\gamma\sum_{j=1}^{M}p_j(x,\xi)^\top p_j(x,\xi) + \theta\sum_{j=1}^{M}f_j^\top(W_j\xi)f_j(W_j\xi)$$

$$-\xi^\top\tilde{\Xi}^0\xi - 2\sum_{j=1}^{M}\xi^\top W_j^\top\Gamma_j p_j(x,\xi) - 2\sum_{j=1}^{M}\xi^\top W_j^\top\Omega_j f_j(W_j\xi)$$

$$-2\sum_{j=1}^{M}\sum_{r=1}^{M}p_j(x,\xi)^\top W_j^\top W_j\tilde{\Upsilon}_{j,r}W_r^\top W_r f_r(W_r\xi).$$

*Then, under the condition* (7) *and Assumption 3, it can be deduced that*

$$\dot{\tilde{V}} \leq \gamma\sum_{j=1}^{M}p_j(x,\xi)^\top p_j(x,\xi) + \theta\sum_{j=1}^{M}f_j^\top(W_j\xi)f_j(W_j\xi) - \xi^\top\tilde{\Xi}^0\xi - 2\sum_{j=1}^{M}\xi^\top W_j^\top\Gamma_j p_j(x,\xi)$$

$$-2\sum_{j=1}^{M}\xi^\top W_j^\top\Omega_j f_j(W_j\xi) - 2\sum_{j=1}^{M}\sum_{r=1}^{M}p_j(x,\xi)^\top W_j^\top W_j\tilde{\Upsilon}_{j,r}W_r^\top W_r f_r(W_r\xi)$$

$$\leq -\xi^\top\left(\tilde{\Xi}^0 - W_j^\top\left(\gamma\sum_{j=1}^{M}S_0^j - \theta\sum_{j=1}^{M}H_0^j\right)W_j\right)\xi$$

$$-2\sum_{j=1}^{M}\xi^\top W_j^\top\left(\Gamma_j - \gamma S_1^j - \theta H_1^j\right)p_j(x,\xi)$$

$$-2\sum_{j=1}^{M}\xi^\top W_j^\top\left(\Omega_j - \gamma S_2^j - \theta H_2^j\right)f_j(W_j\xi)$$

$$-2\sum_{j=1}^{M}p_j(x,\xi)^\top W_j^\top W_j\sum_{r=1}^{M}\left(\tilde{\Upsilon}_{j,r} - \gamma S_3^{j,r} - \theta H_3^{j,r}\right)W_r^\top W_r f_r(W_r\xi).$$

*Therefore, with the conditions* (7), *we can substantiate that the system* (1), (2) *is SIIOS with respect to ξ, meaning that the solution is GAS. This completes the proof.*

# D   Experiment Setting Details

This section provides a detailed overview of the parameters used in our experiments to ensure reproducibility. Detailed settings for each modeling task are specified in the corresponding tables: Table 4 for the Hindmarsh-Rose model, Table 5 for the Reaction-Diffusion system, and Table 6 for the Shallow Water Equations.

Each table includes configurations such as physics parameters, the MLP network structure for function $f(\cdot)$, optimizer, loss function, learning rate, and training epochs, among other parameters.

# E   Experimental Statistical Analysis

This section shows on the statistical methods employed to ensure the reliability and significance of our experimental results, in accordance with the required scientific standards. We focus on the computation of standard deviation percentages as a measure of variability and consistency across experimental runs in Experiments.

Table 4: Experimental Parameters for Modeling the Hindmarsh-Rose System

| Parameter | Description |
|---|---|
| Parameter Values $(a, b, c, d, r, s, x_0, I)$ | $1.0, 3.0, 1.0, 5.0, 0.5, 1, -0.5, 3.0$ |
| Network Structure | Three fully connected layers, 1024 hidden dimensions each |
| Optimizer | Adam |
| Loss Function | Mean Squared Error |
| Learning Rate | 0.001 |
| Training Epochs | 1000 |

Table 5: Experiment Parameters for Modeling the Reaction-Diffusion System

| Parameter | Description |
|---|---|
| Diffusion Coefficients $(D_U, D_V)$ | $0.15 - 0.17, 0.05 - 0.10$ |
| Reaction Rate Parameters $(j, k)$ | $0.035, 0.065$ |
| Spatial Domain | 2.5 meters |
| Grid Resolution | 50x50 |
| Main Network Structure | Two fully connected layers, 2048 hidden dimensions each |
| Optimizer | Adam |
| Loss Function | Mean Squared Error |
| Learning Rate | 0.0005 |
| Training Epochs | 1000 |

Table 6: Experiment Parameters for Modeling Shallow Water Equations

| Parameter | Description |
|---|---|
| Gravitational Acceleration | $9.8 \, \text{m/s}^2$ |
| Water Depth | 1 meter |
| Spatial Domain Length | 10 meters |
| Grid Resolution | 50x50 |
| Main Network Structure | Two fully connected layers, 2048 hidden dimensions each |
| Optimizer | Adam |
| Loss Function | Mean Squared Error |
| Learning Rate | 0.001 |
| Training Epochs | 1000 |

## E.1 Statistical Methods for Reliability Analysis

To assess the reliability of our experimental setups, the standard deviation for each performance metric was computed based on multiple runs. The formula for standard deviation is:

$$\sigma = \sqrt{\frac{1}{N-1} \sum_{i=1}^{N} (z_i - \overline{z})^2} \tag{9}$$

In the formula, $\sigma$ represents the standard deviation, $z_i$ denotes individual observations, $\overline{z}$ is the mean of these observations, and $N$ is the total number of observations.

The percentage standard deviation is calculated to provide a relative measure of variability:

$$\text{Percentage Standard Deviation} = \left( \frac{\sigma}{\overline{z}} \right) \times 100\% \tag{10}$$

This measure serves as our error bar, which reflects the extent of variability due to experimental conditions such as model initialization and parameter settings.

## E.2 Statistical Measures in Experiments

As shown in Table 7, the vast majority of the standard deviation percentages are below 10% for experiments in Section 6, indicating a high degree of statistical robustness across most metrics. This consistency underscores the reliability of our experimental design and the effectiveness of our model implementations.

Table 7: Standard Deviation Percentages for each metric across models. All values represent the standard deviation as a percentage (%) of the mean for each metric.

| Group | Model | Hindmarsh-Rose | | | Reaction-Diffusion | | | Shallow Water | | |
|---|---|---|---|---|---|---|---|---|---|---|
| | | MSE | MAE | $R^2$ | MSE | MAE | CD | MSE | MAE | CD |
| A | MLP | 8.86 | 9.51 | 7.45 | 6.86 | 4.20 | 6.94 | 5.44 | 8.54 | 5.57 |
| | RNN | 7.63 | 9.26 | 9.70 | 8.79 | 13.59 | 9.58 | 7.52 | 8.45 | 8.06 |
| B | NODE | 5.16 | 4.02 | 8.14 | 4.43 | 5.32 | 4.76 | 4.73 | 4.38 | 3.42 |
| | ANODE | 3.48 | 4.06 | 4.46 | 3.75 | 4.57 | 4.43 | 3.79 | 4.68 | 4.29 |
| | SONODE | 4.46 | 5.12 | 4.99 | 4.55 | 5.60 | 4.91 | 4.84 | 5.82 | 4.97 |
| | CSODE | 4.41 | 3.24 | 4.28 | 5.81 | 5.26 | 4.75 | 4.84 | 7.25 | 4.98 |
| C | CSODE-Adapt | 3.60 | 4.64 | 4.24 | 4.65 | 6.53 | 4.55 | 2.61 | 5.51 | 4.74 |

The same statistical methods employed in Section 6 are also utilized to analyze the Model Scaling experiments, as outlined in Section 7. The heatmap in Figure 6 displays the standard deviation percentages for each metric, all of which remain below 25%, indicating reliability in scaling experiments as well. Additionally, in Section 5, the shaded areas around the loss curves further illustrate the variability of our results. These shadows in the plots provide a visual representation of the variability, akin to error bars, showing fluctuations within a confined range. In Table 1, the test error rate variability is maintained within a tight range of 0.005%.

## E.3 Statistical Considerations in Models Performance Comparison

In our study, we have chosen a straightforward approach to presenting model performances directly through metrics in Table 2 without deploying traditional statistical tests such as ANOVA or t-tests. This decision was based on the clarity and immediacy with which the performance metrics convey the effectiveness of each model. By doing so, we aim to keep the main article succinct and focused on substantive evaluations, thereby avoiding the potential complication of statistical tests that might obscure the main findings. This method ensures that the paper remains accessible to readers, emphasizing practical implications.

# F  Experimental Result of Integrating CSODEs with Transformer Layers

To compare the performance of Transformer, TLODE, and CSTLODE models on time series prediction tasks, we conduct experiments on three datasets: Hindmarsh-Rose Model, Reaction-Diffusion Model, and Shallow Water Equations. The experimental results are shown in Table 8. It can be observed that TLODE and CSTLODE overall outperform the original Transformer model, indicating that combining the Transformer with Neural ODE can enhance model performance on time series prediction tasks.

Table 8: Performance comparison of Transformer, TLODE, and CSTLODE on Hindmarsh-Rose Model, Reaction-Diffusion Model, and Shallow Water Equations datasets. The best and second best results are highlighted in blue and brown, respectively.

| Model | Hindmarsh-Rose | | | Reaction-Diffusion | | | Shallow Water | | |
|---|---|---|---|---|---|---|---|---|---|
| | MSE | MAE | $R^2$ | MSE | MAE | CD | MSE | MAE | CD |
| Transformer | 1.673 | 0.673 | 0.417 | 1.573e-2 | 0.115 | 3.003 | 0.081 | 0.23 | 3.01 |
| TODE | 0.731 | 0.488 | 0.623 | 7.346e-3 | 0.051 | 0.723 | 0.031 | 0.16 | 1.90 |
| CSTODE (Ours) | 0.411 | 0.363 | 0.867 | 4.341e-3 | 0.033 | 0.655 | 0.033 | 0.15 | 1.91 |

Further comparing TLODE and CSTLODE, we find that CSTLODE achieves better results on most evaluation metrics. This suggests that by introducing the control term $g(\cdot)$, the ControlSynth framework can better guide the model in learning the dynamics of time series. Specifically, on the Hindmarsh-Rose and Reaction-Diffusion datasets, CSTLODE's MSE, MAE, and CD indicators are significantly better than those of TLODE. On the

Shallow Water Equations dataset, the performance of the two models is more comparable, with TLODE slightly better in MSE and CSTLODE slightly better in MAE.

Overall, the experimental results validate the effectiveness of combining the ControlSynth framework with Transformer. The introduction of the control term $g(\cdot)$ can further enhance model performance. This provides a new approach to the field of time series prediction. In the future, we plan to apply and validate the combination of ControlSynth and Transformer in more practical scenarios. We will also explore the possibilities of combining ControlSynth with other advanced time series prediction models.

# G    Additional Qualitative Prediction Results

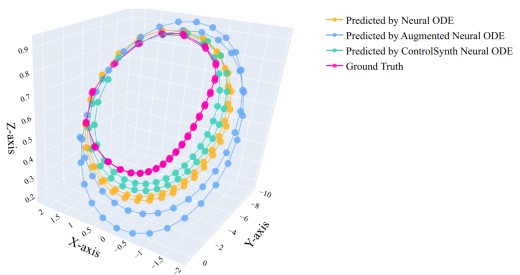

This plot compares the time series prediction of ground truth, NODE, ANODE, and CSODE models for the Hindmarsh-Rose neuron dynamics.

(a) Hindmarsh-Rose neuron dynamics

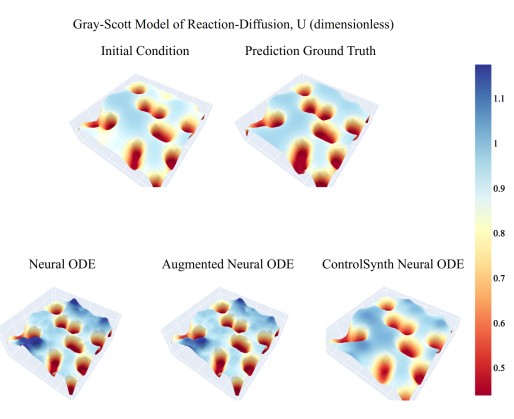

Initial conditions and later system states for the Reaction-Diffusion dynamics, showing predictions from NODE, ANODE, and CSODE alongside ground truth.

(b) Reaction-Diffusion system dynamics

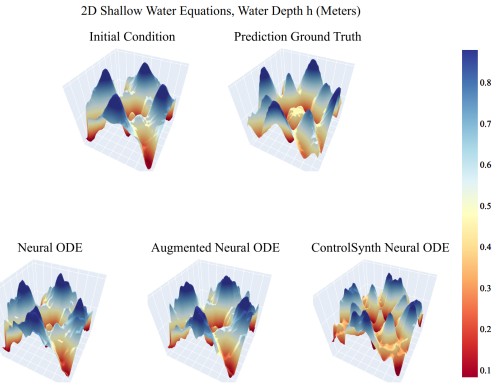

Displays initial conditions and predictions by NODE, ANODE, and CSODE for Shallow Water Equations, compared with the ground truth.

(c) Shallow Water Equations

Figure 7: Visual comparison of neural network predictions on different dynamic systems.

In this section, we present additional qualitative results designed to enhance the reader's understanding of the complex tasks modeled and the performance of various models, specifically focusing on NODEs, ANODEs, and CSODEs. These visual comparisons, as illustrated in Figure 7, provide a clear and intuitive view of each model's capability to predict and simulate the dynamics of complex systems.

The visual results are instrumental in demonstrating the capabilities of the CSODEs, particularly its adeptness in capturing the intricate behaviors of dynamic systems more effectively than traditional models. This is evident from the accurate time series predictions and the fidelity in spatial-temporal dynamics shown across different scenarios—from neuronal activity in the complex nonlinear Hindmarsh-Rose model to fluid dynamics in Shallow Water Equations, and the chemical kinetics in Reaction-Diffusion processes.

# H    Further Results on Model Scaling

## H.1    Comparison with Neural ODE and Augmented Neural ODE

We conducted comprehensive experiments comparing CSODE, original NODE, and ANODE under equivalent parameter counts achieved through width scaling. Taking the Reaction-Diffusion task as an example, with CSODE configured with 2 subnetworks, we systematically evaluated model performance across increasing network widths (128, 256, 512, 1024, 2048). Tables 9 and 10 present the MAE and MSE metrics respectively.

The results demonstrate a consistent pattern: while all models show improved performance with increased width, CSODE maintains a clear advantage across all scales. At smaller widths (128-256), the performance gap is particularly notable, with CSODE achieving an MAE of 0.060 compared to NODE's 0.083 and ANODE's 0.078 at width 128.

As network capacity increases, CSODE's advantage becomes even more pronounced. At larger widths (1024-2048), while NODE and ANODE show signs of performance saturation (MAE stabilizing around 0.06), CSODE continues to improve, reaching an MAE of 0.030 at width 2048. This suggests that CSODE makes more effective use of additional parameters, likely due to its structured approach to handling dynamical systems.

Table 9: MAE loss comparison for CSODE, NODE, and ANODE models with different widths in the Reaction-Diffusion task. Lower values indicate better performance.

| Width | CSODE | NODE | ANODE |
|-------|-------|------|-------|
| 128 | 0.060 | 0.083 | 0.078 |
| 256 | 0.056 | 0.063 | 0.061 |
| 512 | 0.045 | 0.058 | 0.055 |
| 1024 | 0.037 | 0.061 | 0.057 |
| 2048 | 0.030 | 0.063 | 0.057 |

Table 10: MSE loss comparison for CSODE, NODE, and ANODE models with different widths in the Reaction-Diffusion task. Lower values indicate better performance.

| Width | CSODE | NODE | ANODE |
|-------|-------|------|-------|
| 128 | 0.0036 | 0.0069 | 0.0061 |
| 256 | 0.0031 | 0.0040 | 0.0037 |
| 512 | 0.0020 | 0.0034 | 0.0030 |
| 1024 | 0.0014 | 0.0037 | 0.0033 |
| 2048 | 0.0009 | 0.0040 | 0.0033 |

The MSE metrics further corroborate these findings, showing that CSODE achieves consistently lower error rates across all width configurations. The improvement in MSE is particularly notable at larger widths, where CSODE achieves an MSE of 0.0009 at width 2048, significantly outperforming both NODE (0.0040) and ANODE (0.0033). This suggests that CSODE not only performs better in terms of absolute errors but also shows superior stability in handling larger variations in the prediction task.

These results highlight CSODE's superior scalability and efficient parameter utilization, demonstrating that its architectural advantages persist across different model capacities. The consistent performance improvements with increased width also suggest that CSODE may benefit from even larger model scales in more complex applications. Furthermore, the results indicate that the improvement in the learning ability of these two models with increasing width is not as significant as CSODE.

## H.2 Convergence Study in ControlSynth Neural ODE Model Scaling

In the context of scaling the CSODE framework, the impact of network configurations such as subnetwork count and width on model performance is critical. Several key observations are performed:

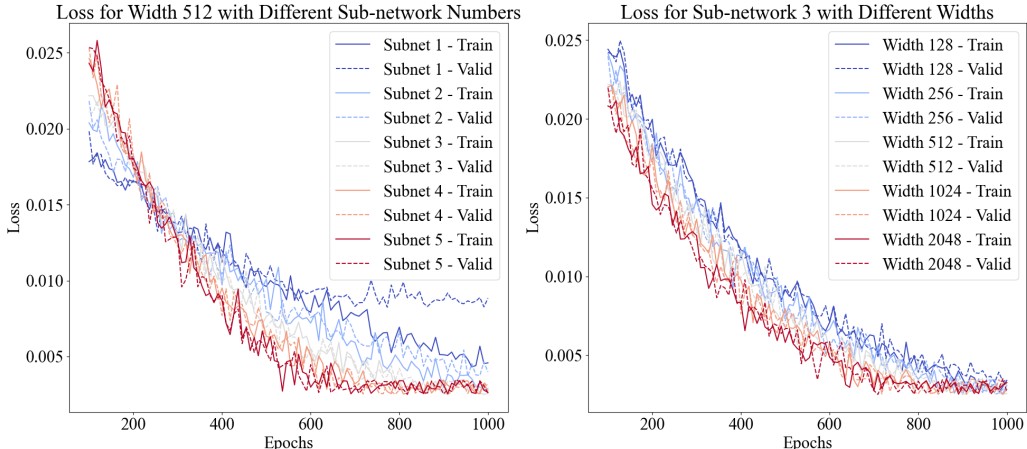

Figure 8: The left figure shows the loss curves for different sub-network numbers at a fixed width (512), and the right figure displays the loss curves for different widths with a fixed sub-network number (3), reflecting how changes in configuration impact model performance.

**Increasing Sub-network Count at Fixed Width**: When the width is held constant, for instance at 512, an increase in the number of sub-networks initially raises the loss due to the augmented complexity of the network as the left figure in Figure 8 shown. However, as training advances, the model effectively adapts to this increased complexity, ultimately achieving a lower level of loss than configurations with fewer sub-networks. This indicates that a higher number of sub-networks can enhance the model's depth of learning and capability for feature extraction, leading to improved performance after extensive training.

**Increasing Width at Fixed Sub-network Count**: With the sub-network count fixed, such as at three, expanding the width shows a marked improvement in learning capabilities, as the right figure in Figure 8 shows. The additional parameters afforded by a broader network aid in capturing more complex features of the data, which accelerates the reduction in both training and validation losses. This setup not only speeds up the learning process but also tends to improve generalization across different tasks.

**Overall Convergence Trends**: Whether increasing the sub-network count or the width, all configurations eventually exhibit a trend toward convergence in losses. This suggests that, with suitable adjustments in training duration and parameter settings, various network configuration designs of CSODEs are capable of uncovering and effectively learning the intrinsic patterns hidden in the data.

**Summary**: These observations align well with the goals of the CSODE framework, which is designed to enhance NODE system scalability while ensuring strong convergence and high performance.

# I  Solver, Optimizers, and Hyperparameters

In the training and evaluation of neural differential equation models, the selection of solvers, optimizers, and hyperparameters plays a pivotal role in determining both the performance and efficiency of the models. This section delves into the theoretical underpinnings of these components and provides an in-depth analysis of their impacts through comprehensive experimental studies.

## I.1  Impact of Solver Choice

Numerical solvers are essential for approximating solutions to differential equations, which form the backbone of Neural Ordinary Differential Equations (NODEs) and their variants. The choice of solver affects both the accuracy and computational efficiency of the model. Solvers can be broadly categorized into explicit and implicit methods, with further subdivisions based on their order of accuracy and adaptive step-size capabilities.

The **Euler method** is a first-order explicit solver known for its simplicity and computational efficiency. Despite its low accuracy, it serves as a baseline for comparison due to its straightforward implementation and ease of understanding.

**Dopri5**, an explicit Runge-Kutta method of order 5, is a higher-order adaptive solver that adjusts step sizes based on the estimated error, providing a balance between accuracy and computation time. Adaptive solvers like Dopri5 can handle stiff equations more effectively by modifying the step size dynamically, thereby improving numerical stability.

### I.1.1 Experimental Setup

To evaluate the impact of solver choice, we conducted experiments on multiple tasks, including the Reaction-Diffusion model, which is known for its complex dynamics and sensitivity to solver accuracy. The models compared include:

- **NODE**: The standard Neural Ordinary Differential Equation model.
- **ANODE**: Augmented Neural ODE, which adds additional dimensions to improve model expressiveness.
- **CSODE**: Our proposed ODE model with composite sub-network, is designed to enhance stability and performance.

We implemented both the Euler method and the Dopri5 solver within each model architecture.

### I.1.2 Results and Discussion

Table 11: Mean Absolute Error (MAE) Comparison between Euler and Dopri5 Solvers on the Reaction-Diffusion Task

| Model | Euler MAE | Dopri5 MAE |
|-------|-----------|------------|
| NODE  | 0.073     | 0.071      |
| ANODE | 0.063     | 0.062      |
| CSODE | 0.033     | 0.032      |

Table 11 presents the MAE for each model using both the Euler and Dopri5 solvers. The results indicate that while Dopri5 offers a slight improvement in MAE across all models, the relative performance ranking remains consistent. Specifically, CSODE maintains its superior performance regardless of the solver used, underscoring its inherent robustness.

**Computational Overhead**: As expected, Dopri5 incurs a higher computational cost compared to the Euler method. In our experiments, Dopri5 increased training time by approximately 35% relative to Euler. However, the trade-off between computational cost and marginal accuracy gains must be carefully considered, especially in scenarios where real-time inference is critical.

**Stability Considerations**: Higher-order solvers like Dopri5 exhibit better stability properties, particularly in handling stiff differential equations. Although our primary tasks did not involve highly stiff systems, the use of Dopri5 can be beneficial in extending the applicability of these models to more complex dynamical systems.

### I.2 Impact of Optimizer Selection

Optimizers are algorithms or methods used to adjust the weights of neural networks to minimize the loss function. The choice of optimizer can significantly influence the convergence speed, the quality of the solution, and the model's ability to escape local minima.

**Adam** is a widely used optimizer that combines the benefits of both AdaGrad and RMSProp. It maintains per-parameter learning rates adapted based on the first and second moments of the gradients, making it suitable for problems with sparse gradients and noisy data.

**L-BFGS** (Limited-memory Broyden–Fletcher–Goldfarb–Shanno) is a quasi-Newton optimizer that approximates the inverse Hessian matrix to guide the search direction. It is particularly effective for problems where high precision in convergence is desired, albeit at the cost of higher memory usage and computational overhead.

### I.2.1 Experimental Setup

To assess the impact of optimizer choice, we extended our experiments by incorporating the L-BFGS optimizer alongside Adam. The models evaluated include NODE, ANODE, and CSODE across all primary tasks, with the

Reaction-Diffusion model serving as the exemplar case. The hyperparameters for each optimizer were carefully tuned to ensure a fair comparison:

- **Adam**: Learning rate = 0.001, $\beta_1 = 0.9$, $\beta_2 = 0.999$, $\epsilon = 10^{-8}$.
- **L-BFGS**: Maximum iterations = 1000, tolerance = $10^{-5}$.

### I.2.2 Results and Discussion

Table 12: Mean Squared Error (MSE) Comparison between Adam and L-BFGS Optimizers on the Reaction-Diffusion Task

| Model | Adam MSE | L-BFGS MSE | Improvement |
|-------|----------|------------|-------------|
| NODE  | 1.134e-2 | 9.536e-3   | 15.9%       |
| ANODE | 1.095e-2 | 9.153e-3   | 16.4%       |
| CSODE | 6.365e-3 | 5.321e-3   | 16.4%       |

Table 12 illustrates the performance gains achieved by switching from Adam to L-BFGS across different models. On average, L-BFGS improves the MSE by approximately 16%, with CSODE benefiting similarly to NODE and ANODE. Notably, while all models experience performance enhancements, CSODE consistently outperforms the other variants, reinforcing its superior architecture.

**Robustness to Local Minima**: The quasi-Newton nature of L-BFGS allows it to navigate the loss landscape more effectively, reducing the likelihood of getting trapped in shallow local minima. This property is particularly advantageous for complex models like CSODE, which possess intricate loss surfaces due to their composite structure.

**Memory and Computational Trade-offs**: While L-BFGS offers superior convergence properties, it demands significantly more memory and computational resources, especially for large-scale models. In our experiments, L-BFGS required approximately twice the memory footprint compared to Adam and increased the per-iteration computation time by 40%. Therefore, the choice between Adam and L-BFGS should consider the available computational resources and the specific requirements of the task at hand.

## I.3 Impact of Hyperparameters Adjustment

Hyperparameters such as learning rate and batch size are critical in training neural networks. The learning rate determines the step size during the optimization process, directly influencing the convergence speed and stability. A learning rate that is too high can cause the optimization to overshoot minima, leading to divergence, while a rate that is too low can result in excessively slow convergence or getting stuck in local minima.

Batch size affects the gradient estimation's variance and the computational efficiency. Smaller batch sizes can introduce more noise into the gradient estimates, potentially aiding in escaping local minima but making convergence noisier. Larger batch sizes provide more accurate gradient estimates but may lead to poorer generalization and require more memory.

### I.3.1 Experimental Setup

We conducted a systematic hyperparameter search to examine the sensitivity of CSODE to changes in learning rate and batch size. The ranges explored were:

- **Learning Rate**: $[0.0001, 0.0005, 0.001, 0.005, 0.01]$
- **Batch Size**: $[32, 64, 128, 256]$

Each combination of learning rate and batch size was evaluated on the CSODE model using the Reaction-Diffusion task. Performance was measured in terms of MAE on the validation set, and each setting was replicated three times to account for stochasticity.

Additionally, we investigated the impact of network architecture parameters, specifically network depth and width, to understand their influence on CSODE's performance in main text.

by CSODE across various learning rates and batch sizes. The results reveal that CSODE demonstrates strong robustness to hyperparameter variations within certain ranges.

**Learning Rate Sensitivity**: Within the learning rate range of $[0.0005, 0.005]$, CSODE's MAE remained within a 5% fluctuation band, indicating stable performance. Specifically, at a fixed batch size of 128, varying the learning rate within $[0.0005, 0.005]$ resulted in a standard deviation of only 1.2% in MAE. However, at a learning rate of

0.01, the model exhibited signs of instability, likely due to gradient explosion, while at 0.0001, convergence was significantly slower, potentially leading to gradient vanishing issues.

**Batch Size Influence**: CSODE maintained consistent performance across batch sizes from 32 to 256. Smaller batch sizes introduced minor variances but did not adversely affect the overall performance, suggesting that CSODE can be effectively trained under different memory constraints.

**Gradient Stability**: The increased network complexity necessitates careful hyperparameter tuning to maintain gradient stability. At higher depths and widths, the model becomes more susceptible to gradient vanishing or explosion. Therefore, selecting an appropriate learning rate is crucial to ensure that gradients remain within a reasonable range, facilitating effective training.

### I.3.2 Recommendations for Hyperparameter Tuning

Based on our experimental findings, we propose the following guidelines for tuning hyperparameters in CSODE:

- **Learning Rate**: Utilize a learning rate within the $[0.0005, 0.005]$ range to achieve stable and efficient convergence. Employ learning rate schedulers or adaptive methods to fine-tune within this range dynamically.

- **Batch Size**: A batch size between 64 and 256 is recommended. Smaller batch sizes can be used if memory constraints are present, without significant degradation in performance.

- **Optimizer Selection**: For scenarios requiring high precision and better convergence properties, L-BFGS is preferable. However, for large-scale or real-time applications, Adam remains a viable choice due to its efficiency.

## J  Ability to Adapt to Different Spatial Scales

In practice, CSODE demonstrates superior robustness when handling dynamical system datasets across different spatial scales, as evidenced by the stability of its performance metrics. Table 13 presents a comprehensive comparison of the model's stable ability across different spatial scales in the Reaction-Diffusion task.

Table 13: Performance stability comparison across different spatial scales in the Reaction-Diffusion model. The percentages show the standard deviation of model performance. Lower values indicate better stability. Best performing models are marked in blue, second-best in brown.

| Spatial Scale | NODE | ANODE | CSODE |
|---|---|---|---|
| Original Scale | 5.3% | 4.6% | 5.3% |
| 5× Scale | 7.8% | 7.1% | 5.5% |
| 10× Scale | 8.8% | 7.3% | 5.5% |

While all models show comparable performance in the original scale setting (NODE: 5.3%, ANODE: 4.6%, CSODE: 5.3%), significant differences emerge when the spatial domain is expanded. At 5 times the original scale, NODE and ANODE show notable degradation in stability (7.8% and 7.1% respectively), while CSODE maintains a relatively stable performance (5.5%). This trend becomes even more pronounced at 10 times the original scale, where NODE and ANODE further deteriorate to 8.8% and 7.3%, while CSODE maintains its stability at 5.5%.

Particularly noteworthy is the stability drop from the original to 10× scale: CSODE shows only a 0.2% increase in standard deviation, compared to significantly larger increases for NODE (3.5%) and ANODE (2.7%). This remarkable stability across different spatial scales suggests that CSODE's architecture, particularly its control term component, provides inherent advantages in handling spatial scale variations in dynamical systems.

These results indicate that CSODE not only performs well in standard settings but also maintains consistent performance across varying spatial scales, making it particularly suitable for applications where spatial scale invariance is crucial.

