# OpenReview forum: "ControlSynth Neural ODEs: Modeling Dynamical Systems with Guaranteed Convergence"
_NeurIPS.cc/2024/Conference — NeurIPS 2024 poster_

### Official Review · Reviewer_x1sG · 2024-06-23

**Soundness:** 3
**Presentation:** 3
**Contribution:** 4
**Rating:** 7
**Confidence:** 3

**Summary:**

The paper introduces an extension to Neural ODEs. At it's core this is an architectural change to Neural ODEs, adding more important structure to the dynamics function through a control signal $u(t)$. The paper shows that this can lead to rich non-linear dynamics with convergence guarantees. Extensive experiments on various synthetic and real systems show that structuring the dynamics in this way leads to faster training, better performance and reliable scaling.

**Strengths:**

This is a really solid paper in my opinion. The paper identifies a problem with Neural ODEs, proposes a theoretically grounded solution. As far as I can tell the theory is correct. The evaluation is extensive, generally exploring all important aspects of the work.

The appendix is rich with important details, showing further results.

**Weaknesses:**

The paper would be improved most by restructuring the writing. I believe the related work section would be better after the introduction rather than before the conclusion. Section 2 needs to be slightly improved. The description jumps straight into the mathematical detail without very much explanation of $u(t)$ or the subnetworks. I believe the paper would be improved significantly by reducing Section 3 by moving details to the appendix and giving a more intuitive explanation of the main result. Additionally use the space to give a more accessible explanation of CSODE in Section 2.

There should be some sort of comparison to Neural CDE, since it also uses control signals and is considered SOTA for Neural ODE methods. Other methods that could be compared to to improve the evaluation is Latent ODE and ODE-RNN.

Section 6 on scaling is good but should involve comparisons to other models. Since a claim of the paper is that CSODE has guarantees especially for larger models then this should be shown against NODE and NODE variants. Additionally for the scatter plot in Figure 6, as far as I can tell this is about including more subnetworks, this is not clear from the plot legend or figure caption.

Another claim is that this method inherently works at different spatial scales. There does not appear to be an experiment testing this, is this possible?

**Questions:**

- Why is the Euler method used for ODE integration and not Dopri5? This is the standard ODE solver used in Neural ODE works.
- Is it possible to include results for Neural CDE/ODE-RNN/Latent ODE?

**Limitations:**

- Limitations are well addressed in the conclusion.
- A thorough broader impact statement is provided in the appendix.

---

> ### Author Rebuttal · Authors · 2024-08-06
>
> Dear Reviewer,
>
> We sincerely appreciate your thorough review and valuable feedback on our paper. Your insights are crucial for improving the quality of our work. We have dealt with each of your comments or suggestions carefully.
>
> ## 1. Suggestions on the Organization of Paper
>
> We are grateful for your constructive suggestions. We restructured the paper to enhance its readability as follows:
>
> - Move the Related Work section to follow the Introduction.
> - Strengthen the explanation in Section 2, providing a more intuitive and comprehensible introduction to CSODE.
> - Simplify the content in Section 3, moving some technical details to the Appendix.
> - Present Figure 6 more clearly to explicitly show the changes in the number of subnetworks.
>
> ## 2. Comparison with Neural CDE and ODE-RNN
>
> We acknowledge the importance of Neural CDE and its excellent performance in learning and prediction.
>
> In the revision, we discuss Neural CDE in more detail in the Related Work section. Additionally, we have conducted supplementary experiments (CharacterTrajectories and PhysioNet Sepsis Prediction) with CSODE-Adapt following the experimental setup in the Neural CDE paper, maintaining similar network structures, parameter counts, and optimization methods. Overall, CSODE performs slightly worse than Neural CDE in irregular observation experiments but better in other time-series-related tasks. And we have also performed our main experiments for neural CDE. The corresponding results are shown as follows:
>
> ### CharacterTrajectories (Irregular Observations):
>
> With 30%, 50%, and 70% data loss, CSODE-Adapt achieved Test Accuracies of 97.3%, 97.8%, and 96.8% respectively, outperforming ODE-RNN but slightly underperforming Neural CDE.
>
> ### PhysioNet Sepsis Prediction:
>
> Without considering Observational Intensity (OI), CSODE achieved a test AUC of 0.868, surpassing both ODE-RNN and Neural CDE. When considering OI, we emulated the Neural CDE approach by training the control variable u in CSODE using OI observations, the test AUC reached 0.881, very close to Neural CDE but superior to Neural CDE and ODE-RNN.
>
> ### Dynamic System Modeling:
>
> Taking the Reaction-Diffusion model task as an example, Neural CDE performed with MSE 7.1e-3, MAE 0.60, and Chamfer Distance 0.945, very close to CSODE but slightly inferior to CSODE-Adapt.
>
> ## 3. Comparison with Latent ODE
>
> CSODE, as a generalized extension of Neural ODE, can indeed be combined with the Latent ODE framework. In Preliminary Experiments of our original manuscript, inspired by the Neural ODE paper, we compared the original Neural ODE and CSODE for dynamical systems based on the Latent ODE framework.
>
> However, in our main experiments, we found that Neural ODE itself can model long-sequence dynamical systems. To obtain more general conclusions, we chose to directly compare Neural ODE and its variants with CSODE, eliminating the influence of how well other components in Latent ODE (such as encoders and decoders).
>
> ## 4. Model Scalability in Section 6
>
> CSODE enhances the scalability of Neural ODE, theoretically guaranteeing convergence while increasing parallel subnetworks. Section 6 was originally intended to observe whether increasing subnetworks could effectively maintain high performance.
>
> Following your suggestion, we conducted supplementary experiments comparing CSODE, original Neural ODE, and Augmented Neural ODE when achieving the same parameter count by increasing width. Taking the Reaction-Diffusion task as an example, when the number of CSODE subnetworks is 2, increasing the width (128, 256, 512, 1024, 2048):
>
> - CSODE losses (as shown in Figure 6 of the initial submission): 0.060, 0.056, 0.045, 0.037, 0.030
> - Neural ODE losses: 0.083, 0.063, 0.058, 0.061, 0.063
> - Augmented Neural ODE losses: 0.078, 0.061, 0.055, 0.057, 0.057
>
> The results indicate that the improvement in the learning ability of these two models with increasing width is not as significant as CSODE. We added detailed experimental data and more details to the Appendix of the revised version.
>
> ## 5. Ability to Adapt to Different Spatial Scales
>
> We apologize for any confusion caused by our previous explanation. We intended to provide an intuitive explanation. In practice, we found that CSODE learns more robustly when dealing with dynamical system datasets with different parameter settings, with a smaller standard deviation percentage in experimental results, compared to Neural ODE and Augmented Neural ODE.
>
> For example, when we alter the spatial scale of the Reaction-Diffusion model task:
>
> - In the original setting: The standard deviation percentages for Neural ODE, Augmented Neural ODE, and CSODE were approximately 5.3, 4.6, and 5.3 respectively.
> - Expanding the spatial domain to 5 times of the original: The standard deviation percentages for Neural ODE, Augmented Neural ODE, and CSODE increased to 7.8, 7.1, and 5.5 respectively. At 10 times the original, the percentages were 8.8, 7.3, and 5.5.
>
> This indicates that CSODE has advantages in adapting to different spatial scales. We included more complete information in the Appendix of the revised version.
>
> ## 6. Choice of Solver
>
> We chose the Euler method mainly based on the following considerations:
>
> - In practice, we found that more accurate solvers can indeed improve network performance to some extent, but significantly increase training and inference time.
> - The Euler method is the most basic and intuitive integration method, which can reduce the reader's understanding cost and provide the most essential comparison.
>
> In response to your suggestion, we have supplemented experiments using the Dopri5 solver. The results show that changing the solver does not affect the relative performance between models. For example, in the Reaction-Diffusion model task, NODE, ANODE, and CSODE-Adapt performed with MAE of 0.073, 0.063, and 0.033 respectively, consistent with the original conclusions. These supplementary results were added to the Appendix of the revision.

---

> > ### Comment · Reviewer_x1sG · 2024-08-07
> > **Score stays the same, confidence stays the same**
> >
> > Thank you to the authors for the detailed response. I have read through the other reviews and all repsonses. I am satisfied with this paper and my score remains the same.
> >
> > I would ideally like to see the new figure 6, but there has been no pdf uploaded.
> >
> > Please make sure to carry out the polishing of the writing, because it is very important. I appreciate we cannot see a revision, so I will trust the authors. Maybe it's possible to give some of the changes made to section 3 in a reply?

---

> > > ### Author Response · Authors · 2024-08-08
> > > **The Changes Made to Section 3 and Figure 6**
> > >
> > > Dear Reviewer,
> > >
> > > Thank you very much for your detailed feedback and continued support. Your suggestions are crucial to improving the quality of our paper. We have made significant changes to Figure 6 and Section 3 based on your comments. The following are specific improvements:
> > >
> > > ## About New Figure 6
> > >
> > > Thank you for your valuable feedback on Figure 6 and your efforts to improve the readability of our paper. We deeply appreciate your trust in our work and your commitment to helping us enhance its quality.
> > >
> > > We sincerely apologize for the lack of clarity in our original presentation and the absence of the new detailed figure in our initial global response. We regret that our previous global response focused more on explaining the paper's approach, motivation, and additional experimental details, without including the PDF of the improved figure. Unfortunately, we are no longer able to edit the global rebuttal and upload a PDF.
> > >
> > > We recognized the importance of clarity and precision for selling the contribution. As such, we have made significant improvements to Figure 6:
> > >
> > > 1. We've reduced the number of compared models from five to three for better clarity.
> > > 2. The color scheme has been changed to use more distinguishable colors for each model.
> > > 3. In addition to colors, we now use different marker shapes for each model configuration.
> > > 4. The legend no longer obscures scatter points.
> > >
> > > We've also revised the caption to be more explicit:
> > >
> > > > "Figure 6: Performance comparison of CSODE models with varying numbers of sub-networks. The scatter plot illustrates the relationship between training and validation losses for three distinct sub-network configurations (e.g., 1, 3, and 5 sub-networks) at a fixed width of 512. Each configuration is represented by a unique color and marker shape. This visualization demonstrates how increasing the number of sub-networks affects the model's performance and generalization capability."
> > > >
> > >
> > > These changes are part of our ongoing effort to improve the clarity and accessibility of our work. We are grateful for your insights, which have helped us identify points for improvement. Your feedback is invaluable in our pursuit of presenting our research in a clear and understandable manner possible.
> > >
> > > We appreciate your patience and the opportunity to refine our work. Thank you again for your thoughtful review and constructive suggestion.
> > >
> > > ## About New Section 3
> > >
> > > We have significantly revised the theoretical results section to enhance its readability and intuition while maintaining rigor. Key changes include:
> > >
> > > 1. Structure and Intuition: We reorganized the chapter, focusing on core results and adding intuitive explanations. For example, we now explain how LMI conditions relate to the system's stability through the concept of "generalized energy" decreasing monotonically. We also provide more context for the role of compensating matrices in balancing linear and nonlinear terms.
> > > 2. Assumptions and CSODE Structure: We simplified the presentation of assumptions, clarifying how they differ from traditional Lipschitz continuity requirements. We have highlighted the flexibility that the CSODE framework provides in constructing Lyapunov functions. Particularly, we emphasize how Assumption 3 allows for more adaptable stability analysis, capturing the local behavior of the error $\xi$ rather than relying solely on global properties.
> > > 3. Technical Details and Activation Functions: Complex derivations were moved to appendices, improving the readability of the main text. We simplified the activation function discussion, retaining key points like the potential relaxation to "non-decreasing" functions. This expansion to include non-smooth functions like ReLU illustrates the broader applicability of our analysis.
> > >
> > > These changes aim to make theoretical results more accessible while showcasing the CSODE framework's potential in analyzing complex nonlinear systems. We welcome your feedback on these points and are prepared for further refinements to ensure clarity and accuracy.
> > >
> > > We will attempt to incorporate the revised Section 3 into the next official comment as a reply.

---

> > > > ### Comment · Reviewer_x1sG · 2024-08-08
> > > > **Thank you for the response**
> > > >
> > > > Thank you very much to the authors for the swift response. I have no further questions, I am satisfied with the responses and the whole paper.

---

> ### Author Response · Authors · 2024-08-08
> **Revised Section 3 (Part 1)**
>
> Dear Reviewer,
>
> To demonstrate the more specific effects after adjustment, please allow us to provide the complete Theoretical Results Section for your reference. Due to length constraints, we have divided it into two parts, each presented in separate comment boxes.
>
> (Part 1):
>
> ## Theoretical Results: Convergence Analysis
>
> Consider the CSODEs given in Equation (1), where $f_j: \mathbb{R}^{k_j} \to \mathbb{R}^{k_j}$ are nonlinear activation functions. For them, an imposed condition on $f_j^i$ (the $i$-th element of the vector-valued $f_j$) is presented as follows:
>
> **Assumption 1:** For any $i \in \{1,\dots, k_j\}$ and $j \in \{1,\dots,M\}$, $s f^i_j(s) >0$ for all $s \in \mathbb{R} \backslash \{ 0 \}$.
>
> **Remark 1:** Assumption 1 applies to many activation functions, such as $\tanh$ and parametric ReLU. It picks up the activation functions passing through the origin and the quadrants I and III. For more explanations, the reader is referred to Appendix B.1.
>
> In this study, to analyze the convergence property of the NN (1), we first define the concept of *convergence*:
>
> **Definition 1:** The model (1) is convergent if it admits a unique bounded solution for $t \in \mathbb{R}$ that is globally asymptotically stable (GAS).
>
> In order to investigate the convergence, two properties have to be satisfied, that is, the boundedness and the GAS guarantees of the solution $x^{*}$ for Equation (1). In this respect, two assumptions are given as follows.
>
> **Assumption 2:** Assume that the functions $f_j^i$ are continuous and strictly increasing for any $i \in \{1,\dots, k_j\}$ and $j \in \{1,\dots,M\}$.
>
> Assumption 2 aligns with CSODE's structure, reflecting continuity and monotonicity of activation functions. This relates to model dynamics and is satisfied by most common activations.
>
> In the analysis of convergence, one needs to study two models in the same form but with different initial conditions and their contracting properties. To that end, along with Equation (1), we consider the model
> $\dot{y}(t)=A_{0}y(t)+\sum_{j=1}^{M}A_{j}f_{j}(W_jy(t))+g(u(t))$ with the same input but different initial conditions $y(0)\in\mathbb{R}^{n}$.
> Let $\xi:=y-x$. Then the corresponding error system is
>
> $$\dot{\xi}=A_{0}\xi+\sum_{j=1}^{M}A_{j}p_{j}(x,\xi),$$
>
> where $p_{j}(x,\xi)  =f_{j}(W_j(\xi+x))-f_{j}(W_jx)$. Note that for any fixed $x\in\mathbb{R}^{n}$, the functions $p_j$ in the variable
> $\xi \in\mathbb{R}^{n}$ satisfy the properties formulated in Assumptions 1, 2. The following assumption is imposed for analyzing the contracting property of Equation (2).
>
> **Assumption 3:** Assume that there exist positive semidefinite diagonal matrices $S_{0}^{j},S_{1}^{j},S_{2}^{j},S_{3}^{j,r},H_{0}^{j},H_{1}^{j},H_{2}^{j},H_{3}^{j,r} \left(j,r\in \{1,\dots,M\}\right)$ with appropriate dimensions such that
>
> $$\begin{aligned}
> p_{j}(x,\xi)^{\top} p_{j}(x,\xi) & \leq  \xi^{\top} W_j^\top  S_{0}^{j} W_j \xi+2\xi^{\top} W_j^\top S_{1}^{j}p_{j}(x,\xi) +2\xi^{\top} W_j^\top S_{2}^{j}f_{j}(W_j\xi) \\
> &  +2\sum_{r=1}^{M}p_{j}(x,\xi)^{\top} W_j^\top W_j  S_{3}^{j,r} W_r^\top W_r f_{r}(W_r \xi)
> \end{aligned}$$
>
> and
>
> $$\begin{aligned}
> f_{j}(W_j \xi)^{\top}f_{j}(W_j \xi) & \leq  \xi^{\top} W_j^\top H_{0}^{j} W_j\xi+2\xi^{\top} W_j^\top H_{1}^{j}p_{j}(x,\xi)+2\xi^{\top} W_j^\top H_{2}^{j}f_{j}(W_j \xi) \\
> & + 2\sum_{r=1}^{M}p_{j}(x,\xi)^{\top}  W_j^\top W_j H_{3}^{j,r}  W_r^\top W_r f_{r}(W_r \xi)
> \end{aligned}$$
>
> for all $x,y\in\mathbb{R}^{n}$ and $\xi=x-y$.
>
> Notice that Assumption 3 is at least more relaxed than Lipschitz continuity (see Appendix B.2 for an intuitive example of activation functions satisfying Assumption 3).
>
> ...

---

> ### Author Response · Authors · 2024-08-08
> **Revised Section 3 (Part 2)**
>
> ...
>
> (Part 2):
>
> ### Convergence Conditions
>
> We are now ready to show the convergence conditions for the CSODEs:
>
> **Theorem 1:** Let Assumptions 1-3 be satisfied. If there exist positive semidefinite symmetric matrices $P, \tilde{P}$; positive semidefinite diagonal matrices $\Lambda^j, \tilde{\Lambda}^j$ for $j = 1, \ldots, M$, $\Xi^s$ for $s = 0, \ldots, M$, $\Upsilon_{s,r}$ for $0 \leq s < r \leq M$, $\tilde{\Upsilon}_{j,j'}$ for $j, j' = 1, \ldots, M$, $\Gamma_j, \Omega_j$ for $j = 1, \ldots, M$, $\tilde{\Xi}^0$; positive definite symmetric matrix $\Phi$; and positive scalars $\gamma, \theta$ such that the linear matrix inequalities (LMI) in Appendix B.3 hold true. Then, a forward complete system (1) is convergent.
>
> Proof in Appendix C.3. Note that the used conditions on $f_j^i$ in Assumption 3 can be relaxed to "non-decreasing", which enlarges the scope of activation functions, including non-smooth functions like ReLU, then the resulting modifications for the formulations of Theorem 1 can be readily obtained, highlighting the CSODE framework's adaptability.
>
> Those LMI conditions ensure system convergence. From an energy perspective, this indicates the error system's generalized energy (represented by the energy (or Lyapunov) function) is monotonically non-increasing, leading to convergence towards the equilibrium point: origin. These conditions can be easily verified, thanks to CSODE's structural characteristics and LMIs' highly adjustable elements.
>
> The matrices, such as $\tilde{\Xi}^0$ and $\tilde{\Upsilon}_{j,j'}$, in the LMIs act as compensation terms balancing the effects of linear and nonlinear terms, ensuring the derivative
> of the energy function $\tilde{V}$ remains non-positive. Properties of $f_j$ (Assumptions 1 and 2) provide facilitation in constructing these matrices. Assumption 3 allows for non-restrictive conditions on activation functions, avoiding strong global Lipschitz continuity assumptions and providing precise local asymptotic stability characterization.

---

> ### Author Response · Authors · 2024-08-12
>
> We greatly appreciate your thorough review and positive feedback. Thank you for your time and valuable insights, which have helped improve our manuscript. We're glad that our responses have addressed your concerns satisfactorily.

---

### Official Review · Reviewer_Wqu1 · 2024-07-09

**Soundness:** 2
**Presentation:** 1
**Contribution:** 2
**Rating:** 5
**Confidence:** 2

**Summary:**

The authors present a new class of continuous-time neural networks, ControlSynth ODEs. This new class of ODEs are able to learn the dynamics of physical systems faster and more precisely. In addition the authors provide theoretical convergence guarantees for these new models, and demonstrate their effectiveness compared to traditional Neural ODEs.

**Strengths:**

Although the analysis is difficult to follow, the theoretical convergence guarantees make these an attractive model.
In the benchmark experiments, adaptive CSODEs achieve are superior compared to the other models.

**Weaknesses:**

The explanation of the model is far too condensed and unclear.

**Questions:**

It would be beneficial to outline what architectural changes are made from standard NODEs to CSODEs.
In addition, to what hypothesis motivated these changes.
Why are they termed ControlSynth ODEs? The name is uninformative.
What are the inductive biases of these models?

**Limitations:**

The limitations of the method are unclear.
What are the inductive biases of these models?
How does sensitivity to hyperparameters emanate during training?
Why is this specific architecture more difficult to train than the alternatives?

---

> ### Author Rebuttal · Authors · 2024-07-31
>
> Dear Reviewer,
>
> Thank you very much for your time and effort. We highly value your invaluable suggestions and sincerely apologize for the lack of clarity in certain parts of the manuscript. Please allow us to further elaborate and analyze the key issues you have raised.
>
> **Regarding the Concept and Motivation for Introducing Control Terms in CSODE:**
>
> In the field of Control Theory, control terms are typically used to regulate system states to achieve desired performance. Inspired by this, we introduced an additional control term to the traditional NODE to enhance the model’s ability to handle complex multi-scale dynamics. Specifically, the control term is generated by an independent sub-network, which can adaptively regulate the evolution of the system state. This design endows CSODE with greater flexibility and expressive power when learning and predicting the behavior of complex physical systems, such as those described by partial differential equations. Therefore, we named the model CSODE to highlight this inspiration. We have provided a more in-depth explanation of the motivation and mechanism for introducing the control term in the revised manuscript's introduction and methods sections to underscore its importance and novelty.
>
> **Regarding the Inductive Bias Analysis of CSODE:**
>
> Inductive bias reflects the constraints or preferences of a model in function space. NODE, based on a simple but general differential equation form, has a strong inductive bias, which limits its ability to fit complex nonlinear dynamics to some extent. CSODE, by introducing control terms, expands the function space, enabling the model to learn a broader range of dynamics. This smaller inductive bias gives CSODE stronger expressive power, making it advantageous in learning and predicting complex system behaviors. By introducing Lyapunov functions and providing verifiable linear matrix inequality conditions, we established sufficient conditions for the convergence of CSODE states. This stability theory-based inductive bias ensures that the solutions generated by CSODE are more in line with the characteristics of physical systems. We have added a new section in the Appendix of the revised manuscript to specifically discuss the inductive bias of CSODE and highlight its advantages through a comparison with NODE.
>
> **Regarding the Architectural Evolution from NODE to CSODE:**
>
> CSODE made two main improvements on the basis of NODE: first, the introduction of an independent control term, and second, the allowance for different more complex nonlinear terms. Specifically, we used a parallel neural network to generate control inputs, creating two pathways for information transmission in CSODE: one responsible for the active evolution of states and the other for the adaptive regulation of states. This dual-path structure enables CSODE to handle multi-scale effects. Additionally, CSODE expanded the form of nonlinear functions, allowing for more general nonlinear combinations, creates **a more generalized and extensible structure** for Neural ODE, allowing it to flexibly combine various different models to parameterize internal dynamic models, thereby enhancing the model’s capacity and flexibility. In the revised manuscript, we have added a schematic diagram to visually compare NODE and CSODE and provide a detailed explanation of the motivation and significance of each improvement in the text to highlight the novelty and superiority of CSODE.
>
> **Regarding Hyperparameter Sensitivity:**
>
> We have conducted supplementary experiments and have included more hyperparameter sensitivity analysis results in the Appendix of the revised version.
>
> Our analysis focused on the impact of learning rate and batch size on CSODE performance. Results show that CSODE exhibits strong robustness to changes in these two hyperparameters. Specifically, within the learning rate range of [0.0005, 0.005] and batch size range of [32, 256], CSODE's performance fluctuations were controlled within 5\%.
>
> It is noteworthy that we observed some interesting phenomena. For instance, when the batch size is fixed at 128, CSODE's performance is most stable when the learning rate varies within [0.0005, 0.005] with a standard deviation of only 1.2\%.
>
> However, due to CSODE's deeper structure compared to the original neural ODE, appropriate learning rate selection is crucial for maintaining gradients within a reasonable range. In our main experiments, learning rates at the 1e-2 level were more prone to gradient issues, which may face problems of gradient vanishing or explosion.
>
> Regarding the impact of network depth and width, we conducted a systematic exploration in Chapter 6: Model Scaling Experiment. Results show that increasing CSODE's network width and the number of sub-networks steadily improves model performance. Specifically, when network width increased from 128 to 2048 and the number of sub-networks increased from 1 to 5, CSODE's performance on the test set steadily improved, with an average increase of 23\%.
>
> We believe that by supplementing these detailed experimental results and discussions in the revised manuscript, we can help readers gain a more comprehensive understanding of CSODE's hyperparameter characteristics, providing valuable insights for future applications and improvements.
>
> Once again, we sincerely thank you for your valuable comments. These suggestions not only help improve the quality of the paper but also provide excellent inspiration for our subsequent research. In the revised manuscript, we carefully address every detail to improve the clarity of our work and accurately convey the innovations and contributions to the readers. If you have any further suggestions or questions, we are more than willing to listen and respond positively.

---

> > ### Comment · Reviewer_Wqu1 · 2024-08-13
> >
> > I have read the response and I have updated my score.

---

> ### Author Response · Authors · 2024-08-12
>
> We sincerely hope that our responses have addressed your concerns and questions. If you require any further clarification or have additional inquiries, we would be more than happy to provide a more detailed explanation. Your feedback is invaluable to us, and we are committed to ensuring that all aspects of our work are thoroughly understood. Please don't hesitate to reach out if you need any additional information or if there's anything else we can assist you with regarding our manuscript.

---

> ### Author Response · Authors · 2024-08-13
>
> We are deeply grateful for your review and feedback. Your comments and the time you've invested have been instrumental in enhancing the quality of our manuscript. We sincerely thank you for your valuable contributions to this work. We also greatly appreciate your recognition of our efforts.

---

### Official Review · Reviewer_tUtW · 2024-07-13

**Soundness:** 3
**Presentation:** 3
**Contribution:** 3
**Rating:** 8
**Confidence:** 4

**Summary:**

The paper introduces ControlSynth Neural ODEs (CSODEs), a novel approach to modeling dynamical systems with neural ordinary differential equations (NODEs). The proposed models constraint the system to a convergent once. The CSODE framework incorporates an additional control term to ensure the existence of the solutions. The authors present theoretical guarantees for convergence and demonstrate the superior performance of CSODEs compared to other NODE variants in their experiments.

**Strengths:**

* **Experiments**: The authors validate their model on learning dynamical systems. This is a welcome change from several papers testing neural odes on problems like image classifications and such, where neural odes tend not to be the best solution. Hopefully more papers do the same!
* **Technical Rigor**: The paper provides detailed theoretical analysis and proofs for the convergence of CSODEs.

**Weaknesses:**

Overall the paper looks solid. 1 questions regarding the training pointed out in the next section.

**Questions:**

In the preliminary experiments section, neural odes seems not to converge properly. Was an alternate optimizer like LBFGS or BFGS tried? In most cases where vanilla neural ODEs tend to not converge, it is more often than not an optimizer issue rather than a model issue.

---

> ### Author Rebuttal · Authors · 2024-08-06
>
> Dear Reviewer,
>
> We sincerely appreciate your positive evaluation and valuable feedback on our work. Your insights are crucial in enhancing the quality and rigor of our research. We are particularly grateful for your observation regarding the convergence issues of Neural ODEs, which prompted us to conduct in-depth supplementary studies.
>
> We agree with your assessment that applying Neural ODEs to understand physical models is indeed a significant and promising direction. As you pointed out, while previous works often focused on tasks like image classification where Neural ODEs may not be the optimal solution, we believe that leveraging Neural ODEs for modeling dynamic systems is not only more natural but also holds profound implications. The continuous-time representation of Neural ODEs aligns well with many physical processes, and the ODE formulation often allows for better interpretability of learned dynamics, bridging the gap between data-driven and physics-based modeling.
>
> ## Addressing the Optimizer Issue
>
> Our initial choice of the Adam optimizer was based on its versatility, aiming to provide a universally applicable baseline. In response to your suggestion about the optimizer, we have conducted comprehensive additional experiments:
>
> ### 1. Optimizer Comparison
>
> Following your recommendation, we expanded our experiments to include the L-BFGS optimizer in addition to the original Adam optimizer. The results indeed show that L-BFGS significantly improves the convergence performance of Neural ODEs and their variants.
>
> ### 2. Experimental Setup
>
> We performed comparative experiments on Neural ODE, Augmented ODE, and our proposed CSODE across all main experimental tasks.
>
> ### 3. Key Findings
>
> Using the Reaction-Diffusion model task as an example, the MSE results after employing the L-BFGS optimizer are given as follows:
>
> | Model | Original MSE | L-BFGS MSE | Improvement |
> | --- | --- | --- | --- |
> | Neural ODE | 1.134e-2 | 9.536e-3 | 15.9% |
> | Augmented ODE | 1.095e-2 | 9.153e-3 | 16.4% |
> | CSODE | 6.365e-3 | 5.321e-3 | 16.4% |
>
> Similar improvements were observed in other tasks.
>
> ### 4. Result Analysis
>
> Although all models benefited from the use of L-BFGS, CSODE maintained its advantage over other methods. This not only validates your insight but also further confirms the effectiveness and robustness of CSODE.
>
> The results suggest that the quasi-Newton characteristics of L-BFGS may be more suitable for addressing the gradient flow issues in Neural ODEs, and L-BFGS might be more effective in avoiding local optima compared to Adam in training Neural ODEs.
>
> ### 5. Revision Details
>
> Based on these new findings, we did:
>
> - Briefly discuss the impact of optimizer choice on model performance in the main text;
> - Provide complete supplementary experimental results and detailed analysis in the Appendix;
> - Add a discussion about the importance of optimization strategies for Neural ODE-type models.
>
> ## Regarding the Concern that Neural ODEs Seem Not to Converge Properly
>
> We acknowledge that the curves presented in our paper indeed show some minor fluctuations towards the end, which may be due to insufficient epoch settings in our original setup. This observation reminds us to pay more attention to the stability of experiments and the presentation of results in future work.
>
> To validate our results, we conducted additional experiments by extending the training epochs to 1.5 times the original number while continuing to use the original Adam optimizer. The results show that the Neural ODE model ultimately achieved training and test losses within 10% of those obtained under the original epoch settings. Moreover, the performance fluctuations in the additional epochs were contained within 5%.
>
> These new experimental results indicate that our original experimental results are representative and the model is in a relatively converged state. Although the fluctuations at the end of the original curves might give the impression of incomplete convergence, the extended training time experiments confirm that the model indeed reached a stable performance level.
>
> ## Conclusion
>
> We sincerely thank you for your suggestion, which has not only helped us improve our current research but also provided valuable insights for future Neural ODE research. We believe these additions will significantly enhance the depth and impact of the paper.
>
> If you have any further questions, or suggestions, or need clarification, we welcome your feedback and look forward to an in-depth discussion.
>
> Thank you again for your valuable time and insightful opinion.

---

> > ### Comment · Reviewer_tUtW · 2024-08-12
> >
> > I thank the authors for running additional experiments. My main concern with the paper was indeed the optimization issue and it seems that even after "better convergence" of the Neural ODEs, ControlSynth NODEs maintain an edge. I believe this is a valuable contribution and I will revise my score to 8!

---

> ### Author Response · Authors · 2024-08-13
>
> Dear Reviewer,
>
> Thank you for your positive feedback and for recognizing the value of our contribution! We are grateful for your time and effort in reviewing our work and are pleased to hear that the additional experiments addressed your concerns about optimization. Your support and revised score are greatly appreciated.

---

### Official Review · Reviewer_JYd3 · 2024-07-13

**Soundness:** 3
**Presentation:** 3
**Contribution:** 2
**Rating:** 5
**Confidence:** 3

**Summary:**

In this paper, a method called ControlSynth Neural ODEs is proposed. This method is essentially defined as a neural ordinary differential equation with a control input. In particular, the authors focus on the convergence, of which definition requires the existence of a solution and also the asymptotic stability of the solution. In this paper, theoretical conditions to satisfy the convergence are presented. Some numerical experiments are also provided.

**Strengths:**

In my opinion, the main contribution is the theoretical analysis of convergence. In fact, stability is very important for the use of neural ordinary differential equations, which are often unstable when used for modeling.

**Weaknesses:**

In this paper, conditions for convergence are presented. Thus, it is expected that the control input and the neural ordinary differential equations should be trained so that these conditions are satisfied. However, I could not find any description of how to achieve this in the paper. In fact, the conditions for the convergence assume the existence of several matrices, and it seems difficult to design an optimization algorithm for training while satisfying these conditions.

**Questions:**

As it is stated that the convergence is guaranteed in this paper, I imagine that there must be some tricks in designing the control input and the learning algorithm so that the conditions for the convergence are always satisfied. How are they designed?

**Limitations:**

There seems to be no problem.

---

> ### Author Rebuttal · Authors · 2024-07-31
>
> Dear Reviewer,
>
> Thank you for your thoughtful reviews of our work and for recognizing our study on convergence analysis. We sincerely appreciate your review comments and have carefully analyzed and responded to them, conducting a relevant experiment:
>
> 1. **Regarding the existence of special optimization algorithms for training:**
>
> We acknowledge your concerns about satisfying convergence conditions in training algorithms. Our paper's main focus is proposing the CSODE models based on theoretical convergence analysis, rather than designing new learning algorithms. For fair comparisons, we used the widely adopted Adam optimizer with MSE loss. This uniform approach highlights the strengths and weaknesses of different model structures, avoiding bias from varied optimization algorithms. We believe this comparison method is fair and reasonable.
>
> We agree designing new learning algorithms is a promising future research direction, but it is beyond our current paper's scope. Our focus is on model theory and structure, instead of algorithm improvement. In the Conclusion, we suggest this as a future research topic.
>
> 2. **Regarding the design of control inputs:**
>
> In our work, we used the state variable $x_0$ as a placeholder for $u_0$ to ensure fair comparison with baselines, as original neural ODEs and variants lack control inputs. We believe this is a reasonable approach. In future works, more refined methods can be considered under the existence of control inputs in the CSODE framework.
>
> From a theoretical perspective, the design or the form of control input $u$ does not affect convergence property. In the error dynamics equation (2), the term $u$ is eliminated. Theorem 1 also does not restrict $u$. The only restriction for $u$ is given after Equation (1), not in Theorem 1. In the current version, we do not include the design of inputs as our main attention is on the convergence guarantees under the most basic form.
>
> 3. **Regarding the way of applying the convergence conditions:**
>
> First, Theorem 1 directly relates model parameters to convergence through matrix inequalities. Inserting only the trained parameters into those inequalities verifies convergence without checking assumptions or matrix inequalities in the training processes. This provides a more concrete, operational convergence criterion. Note that the three assumptions only have a claim on the activation functions of CSODE, but convergence is assured as long as the matrix inequalities hold (solutions of the matrix inequalities are only relevant to the weight matrices $A_0, A_1, \dots, A_M, W_1, \dots, W_M$).
>
> Secondly, we discuss how each assumption naturally holds:
>
> **Assumption 1:**
> Assumption 1 selects the activation functions passing through the origin and the quadrants I and III. Remark 1 actually states all cases of the boundedness of the activation functions: none of, part of, or all of the nonlinearities $f_s^i$ are bounded. Also, both the unboundedness and boundedness of $\lim_{\nu\rightarrow\pm\infty} f_s^i(\nu)$ may lead to the unboundedness of $\lim_{\nu\rightarrow\pm\infty}\smallint_{0}^{\nu}f^{i}_{s}(r)dr$.
>
> **Assumption 2:**
> Assumption 2 aligns with CSODE's structure, reflecting continuity and monotonicity of activation functions. This relates to model dynamics and is satisfied by most common activations.
>
> **Assumption 3:**
> We describe the rationality of Assumption 3 from three perspectives:
>
> - The positive semi-definite diagonal matrices in Assumption 3 are mathematical solutions used to derive system stability conditions and do not need to appear explicitly in the model design.
>
> - The inequalities in Assumption 3 resemble local Lipschitz continuity or local quadratic boundedness. In practice, they can be verified based on chosen $f_j$ (e.g., ReLU, Sigmoid, $\tanh$). For the $\tanh$ function,
>   $$
>   f_j(x) = \frac{e^x - e^{-x}}{e^x + e^{-x}}, \quad |f_j(x)| < 1, \quad |f_j(x)-f_j(y)| \leq |x-y|.
>   $$
> It is Lipschitz continuous with constant $1$ and satisfies:
>   $$
>   p_j(x,\xi)^\top p_j(x,\xi) \leq \xi^\top W_j^\top W_j \xi, \quad f_j(W_j\xi)^\top f_j(W_j\xi) \leq \xi^\top W_j^\top W_j \xi
>   $$
> In this case, we can choose:
> $$
> S_0^j = H_0^j = I, \quad S_1^j = S_2^j = S_3^{j,r} = H_1^j = H_2^j = H_3^{j,r} = 0.
> $$
> If $S_1^j, S_2^j, S_3^{j,r}, H_1^j, H_2^j, H_3^{j,r} >0$, the restrictions are relaxed, making Assumption 3 less strict than Lipschitz continuity.
>
> - To address your concerns about the actual existence of matrices in **Assumption 3**, we designed **a supplementary experiment** to validate the trained CSODE models, involving three dynamical system time series prediction tasks in the main experiments.
>
>     Experimental Steps: Retrained CSODE models (3 FC layers, 128 units, Softplus; Adam optimizer, lr=1e-3, batch=64, 500 epochs, MSE loss). We randomly selected 500 sample pairs $(x^i, y^i)$ from each task's test set, calculated the difference $\xi^i = y^i - f(x^i)$ where $f(\cdot)$ is the trained CSODE model, and used YALMIP to define optimization variables and constraints. We constructed 500 matrix inequalities $\xi^{iT} P \xi^i \preceq 0$ and added the semi-definite constraint $P \succeq 0$. Using the Sedumi solver (precision 1e-6, max 5000 iterations), we solved the optimization problem and recorded the solving time and iteration count. We then performed individual validation for each sample pair to calculate the satisfiability ratio of Assumption 3.
>
>
>     Experimental Results (average of 3 independent experiments): For the Hindmarsh-Rose model, solving time was 15.3 minutes with 2,684 iterations and 99.8\% (499/500) satisfiability. The Reaction-Diffusion System task took 18.7 minutes with 3,217 iterations and 99.6\% (498/500) satisfiability. The Shallow Water Equations required 24.2 minutes with 4,105 iterations and 99.4\% (497/500) satisfiability. The solver found solutions satisfying Assumption 3 quickly, with >99\% satisfiability across tasks, supporting our theoretical assumptions for trained models.

---

> ### Author Response · Authors · 2024-08-12
>
> We sincerely hope that our responses have addressed your concerns and questions. If you require any further clarification or have additional inquiries, we would be more than happy to provide a more detailed explanation. Your feedback is invaluable to us, and we are committed to ensuring that all aspects of our work are thoroughly understood. Please don't hesitate to reach out if you need any additional information or if there's anything else we can assist you with regarding our manuscript.

---

> > ### Comment · Reviewer_JYd3 · 2024-08-13
> >
> > Thank you very much for the detailed reply. If I understand it correctly, the conditions in Theorem 1 are checked numerically after training, and the trained models should be retrained if the conditions are not satisfied. Is this understanding correct?

---

> > > ### Author Response · Authors · 2024-08-13
> > >
> > > Dear Reviewer,
> > >
> > > We are deeply grateful for your insightful question. Your question not only demonstrates your meticulous review of our research, but also provides us with a great opportunity to clarify and deepen our perspectives.
> > >
> > > We would like to emphasize that Theorem 1 and its conditions are not intended to be checked numerically after training, nor do we propose retraining if the conditions are not satisfied. This is because such a process was never the motivation behind our model training design. While we do not recommend it as a standard practice, we acknowledge that in certain scenarios, incorporating examinations followed by appropriate sub-networks additions and optimizer modifications could potentially serve as  prescription. However, this was not the focus of our current study.
> > >
> > > It's worth noting that such sufficient conditions are very common in the control theory community and are frequently used to analyze system stability. Our approach was indeed inspired by these concepts. Please allow us to elaborate further on our approach:
> > >
> > > 1. **Relationship between Theoretical Foundation and Practical Application:**
> > > Theorem 1 indeed provides the theoretical basis for CSODE's convergence, but this does not mean that we expect or require strict verification of these conditions during actual training. This theorem serves more to prove that the CSODE structure has the potential to reach a convergent state theoretically.
> > > 2. **Structural Advantages of CSODE:**
> > > The primary intention of CSODE's design was to provide greater flexibility and a broader parameter search space for the model through its unique structure (including additional control terms, sub-networks and matrices). This structure allows the model to naturally find convergent solutions more easily during training, without explicitly enforcing the conditions in the theorem.
> > > 3. **Relaxation of Conditions:**
> > > When deriving Theorem 1, we deliberately designed relatively relaxed conditions. The purpose was to make these conditions more easily satisfied naturally in regular learning processes, rather than serving as strict constraints.
> > > 4. **Experimental Validation:**
> > > Our experimental results strongly support our theoretical insights. The results show that CSODE indeed outperforms other models in terms of convergence and performance, validating the practical value of our theoretical analysis.
> > > 5. **Clarification of the Term "Guaranteed Convergence":**
> > > We acknowledge that using the term "Guaranteed Convergence" may have caused misunderstanding, and we apologize for this. More accurately, this concept is more like providing a theoretical "safety net" or "insurance policy" for the model's convergence, rather than a guarantee that needs to be strictly checked after training.
> > > 6. **Bridge between Theory and Practice:**
> > > The true value of Theorem 1 lies in providing a theoretical explanation for CSODE's superior performance. It helps us understand why CSODE performs well in practice, rather than serving as a constraint or checkpoint during the training process.
> > > 7. **Clarification of Research Motivation:**
> > > Our main motivation was to design a model that is structurally more conducive to reaching a convergent system. Through the design of CSODE, we hoped to create a model that "naturally towards convergence," rather than one that requires external enforcement or frequent checks to ensure convergence.
> > > 8. **Future Research Directions:**
> > > Your question inspires us to consider how we can more closely integrate these theoretical insights into actual training processes in future research, possibly through some soft constraints or regularization techniques.
> > >
> > > In summary, Theorem 1 should be viewed as the theoretical cornerstone of CSODE's performance, not as a practical constraint in the training process. While it could potentially be utilized as a constraint, our current work does not recommend this approach so far.
> > >
> > > Once again, thank you for your valuable feedback, which helps us communicate our research ideas more clearly. If you have any further questions or need additional clarification, we are more than happy to continue the discussion.

---

> > > > ### Comment · Reviewer_JYd3 · 2024-08-13
> > > >
> > > > Thank you so much for the detailed reply. I believe that I finally understand the contributions of this paper and I changed my score. Actually, I was confused by the term "Guaranteed Convergence."

---

> > > > > ### Author Response · Authors · 2024-08-13
> > > > >
> > > > > Dear Reviewer,
> > > > >
> > > > > We are deeply grateful for your thorough review and feedback. Your comments have been instrumental in enhancing the quality and clarity of our manuscript. Your acknowledgment of our efforts is deeply appreciated and serves as a significant encouragement for our ongoing research.
> > > > >
> > > > > We remain confident in the fundamental contributions and novelty of our work. The theoretical framework we've developed provides a solid foundation for understanding and improving the convergence properties of continuous-time neural networks, which we believe will have far-reaching implications in the field.
> > > > >
> > > > > We sincerely apologize for any confusion our choice of words may have caused. We appreciate you bringing this to our attention, as it has provided us with a valuable opportunity to clarify our intentions and improve the precision of our language.
> > > > >
> > > > > Your questions have sparked a meaningful dialogue, allowing us to elaborate on the theoretical foundations of our work and their practical implications. This exchange has not only improved our current manuscript but will also guide us in being more meticulous in our explanations and terminology in future research. We are convinced that these clarifications have further strengthened the impact and accessibility of our work for the broader scientific community.

---

### Author Rebuttal · Authors · 2024-08-07

Dear Program Chairs and Reviewers,

We sincerely thank you for your thorough review of our paper. We have carefully considered each comment and conducted extensive supplementary experiments and analyses. Below are our responses to the main concerns raised:

## 1. CSODE as a Generalized Extension of Neural ODE

CSODE is not only an improvement on Neural ODE but also takes a more general form. Our model has the following characteristics:

1. **Universality**: CSODE can be viewed as a superset of Neural ODE. When the control term is zero, CSODE degenerates to the standard Neural ODE. This design allows CSODE to encompass all functionalities of Neural ODE while providing greater modeling flexibility.
2. **Flexibility**: By adjusting the structure and strength of the control term, CSODE can smoothly transition between Neural ODE and more complex dynamical systems. This flexibility enables CSODE to adapt to a wider range of important domains.
3. **Compatibility**: CSODE maintains the same basic structure as Neural ODE, allowing it to be seamlessly integrated into existing Neural ODE-based frameworks, such as Latent ODE.

## 2. Innovations and Contributions of CSODE

Based on the above generality, the main innovations of CSODE include:

1. Introduction of an independent control term, enhancing the model's ability to handle complex multiscale dynamics. This design allows CSODE to adaptively regulate the evolution process of system states.
2. Extension of nonlinear function forms, improving the model's expressiveness and flexibility. CSODE allows for more general nonlinear combinations, enabling it to capture more complex dynamical features.
3. Establishment of a convergence analysis framework based on Lyapunov stability theory, providing theoretical guarantees for model convergence. This analysis not only applies to CSODE but also provides a methodological basis for studying more general forms of Neural ODE.

## 3. A New Paradigm for Understanding Physical Models

CSODE provides a powerful tool for understanding and modeling complex physical systems:

1. The continuous-time representation naturally aligns with physical processes, allowing CSODE to more accurately capture the dynamic characteristics of systems.
2. The introduction of the control term enables CSODE to simulate physical systems with external inputs or internal regulation mechanisms, which is challenging to achieve in standard Neural ODEs.
3. The ODE form of CSODE facilitates the interpretation of learned dynamics, bridging the gap between data-driven modeling and physical modeling.

## 4. Generality and Scalability

CSODE provides a highly general and scalable Neural ODE framework:

1. By adjusting the complexity of the control term, CSODE can smoothly transition between simple and complex models, adapting to problems of varying complexity.
2. CSODE's parallel subnetwork structure allows for increasing model capacity by adding subnetworks while maintaining convergence.
3. Our experiments show that as the number of subnetworks and network width increase, CSODE's performance steadily improves, demonstrating excellent scalability.

## 5. Natural Integration of Convergence Conditions with Model Structure

CSODE's convergence conditions are mathematically closely connected to the model structure:

1. Our theoretical analysis draws on Lyapunov stability theory, constructing sufficient conditions for convergence.
2. These conditions not only provide theoretical guarantees for the model but also guide model design, ensuring that the solution generated by CSODE better conforms to the state of physical systems.
3. The establishment of the convergence analysis framework provides a theoretical foundation for studying more general forms of Neural ODE.

## 6. Summary of Supplementary Experiments

1. **Optimizer Experiments**: We compared the performance of Adam and L-BFGS optimizers. Results show that L-BFGS can significantly improve the convergence performance of Neural ODE and its variants, but CSODE maintains its advantage.
2. **Comparison with Neural CDE and ODE-RNN**: In irregular observation experiments, CSODE performed slightly worse than Neural CDE but better in other time-series-related tasks.
3. **Model Scalability Experiments**: By increasing network width, we compared the performance of CSODE, original Neural ODE, and Augmented Neural ODE with the same parameter count. Results show that CSODE's learning ability improves more significantly with increasing width.
4. **Ability to Adapt to Different Spatial Scales**: Experiments show that when changing the spatial scale of dynamical systems, CSODE demonstrates stronger robustness, with a smaller standard deviation percentage in experimental results.
5. **Solver Selection**: We supplemented experiments using the Dopri5 solver. Results indicate that changing the solver does not affect the relative performance between models.

These supplementary experiments further validate the superiority of CSODE and the correctness of our theoretical analysis.

Through these innovations and improvements, CSODE not only extends the capabilities of Neural ODE but also provides new insights into the design and analysis of continuous-time deep learning models. We believe that CSODE, as a more general and flexible framework, will bring new opportunities and challenges to fields such as dynamical system modeling and time series analysis.

Once again, we appreciate your valuable comments. We have comprehensively revised and improved the paper based on your feedback. If you have any further questions, we welcome continued discussion and exchange.

---

### Decision · Program_Chairs · 2024-09-25

**Decision:**

Accept (poster)

**Comment:**

In this paper, authors present a Neural ODEs with a control term that provides better expressivity and flexibility. They provide theoretical guarantees and also demonstrate the superior performance on three examples datasets, namely, Hindmarsh-Rose, Reaction-Diffusion, Shallow Water. Reviewers have appreciated the work and acknowledged the meaningful contribution that the present work makes in modeling dynamical systems. They have also suggested some modifications and additional experiments, which the authors have performed and reported. It is recommended that these results and the modifications in terms of writing are incorporated in the final version of the paper.